# STITCH: SIMULTANEOUS THINKING AND TALKING WITH CHUNKED REASONING FOR SPOKEN LANGUAGE MODELS

**Cheng-Han Chiang**[1,2*]  **Xiaofei Wang**[2†]  **Linjie Li**[2]  **Chung-Ching Lin**[2]  **Kevin Lin**[2]
**Shujie Liu**[2]  **Zhendong Wang**[2]  **Zhengyuan Yang**[2]  **Hung-yi Lee**[1]  **Lijuan Wang**[2]

[1]National Taiwan University    [2]Microsoft

## ABSTRACT

Spoken Language Models (SLMs) are designed to take speech inputs and produce spoken responses. However, current SLMs lack the ability to perform an internal, *unspoken* thinking process before responding. In contrast, humans typically engage in complex mental reasoning internally, enabling them to communicate ideas clearly and concisely. Thus, integrating an unspoken thought process into SLMs is highly desirable. While naively generating a complete chain-of-thought (CoT) reasoning before starting to talk can enable thinking for SLMs, this induces additional latency for the speech response, as the CoT reasoning can be arbitrarily long. To solve this issue, we propose **STITCH**, a novel generation method that alternates between the generation of unspoken reasoning chunks and spoken response chunks. Since the audio duration of a chunk of spoken response is much longer than the time to generate the tokens in a chunk of spoken response, we use the remaining free time to generate the unspoken reasoning tokens. When a chunk of audio is played to the user, the model continues to generate the next unspoken reasoning chunk, achieving **simultaneous thinking and talking**. Remarkably, STITCH matches the latency of baselines that cannot generate unspoken CoT by design while outperforming those baselines by 15% on math reasoning datasets; STITCH also performs equally well on non-reasoning datasets as those baseline models. Some animations and demonstrations are on the project page: `https://d223302.github.io/STITCH`.

## 1 INTRODUCTION

Spoken language models (SLMs) can take speech inputs and generate speech outputs (OpenAI, 2024b; Défossez et al., 2024; Arora et al., 2025). SLMs generate speech by emitting some *speech tokens*, discrete tokens that encode the information in speech (Défossez et al., 2023; Zhang et al., 2024), which will be converted into an audio waveform by a speech decoder. Most SLMs are trained from a backbone text-based LLM (Défossez et al., 2024; Ding et al., 2025); we can obtain an SLM by teaching the backbone LLM to understand speech input and generate the speech tokens as the output (Xiezhifei, 2024; Fang et al., 2025a).

Teaching the backbone LLM to directly generate speech tokens is challenging, as the LLM is originally trained to predict text tokens, so the speech tokens are completely new for the LLM backbone. To make SLMs better predict the speech tokens, current mainstream methods first generate the text tokens before generating the speech tokens (Chen et al., 2024; Zeng et al., 2024; Nguyen et al., 2025). By design, those text tokens and the speech tokens are *aligned*, in the sense that the text tokens correspond to the semantic content of the speech tokens.

While the aligned text-speech generation pipeline makes it easier to generate the speech response, these SLMs directly generate the output text-speech tokens without any inner thinking process, though explicitly modeling such reasoning steps has been shown helpful in text generation tasks (OpenAI,

---

*Work done during an internship at Microsoft GenAI. `dcml0714@gmail.com`
†Correspondence: `xiaofei.wang@microsoft.com`

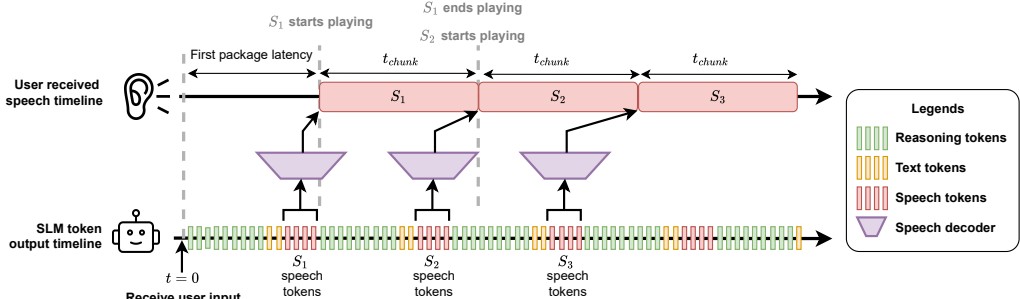

Figure 1: The timing diagram during generation for STITCH-R. The model first generates the first $N_{reason}$ CoT reasoning tokens, $N_{text}$ text tokens, and $N_{speech}$ speech tokens. Once the first $N_{speech}$ speech tokens are generated, the speech decoder synthesizes the output audio that lasts $t_{chunk}$ seconds. When the speech waveform is synthesized and played to the user, the SLM uses this time to generate the next $N_{reason}$ reasoning tokens, $N_{text}$ text tokens, and $N_{speech}$ speech tokens and synthesize the speech output. The duration $t_{chunk}$ is much longer than the time for generating the text tokens and speech tokens corresponding to $S_i$, and we use the remaining time to generate the reasoning tokens.

2024c; Guo et al., 2025). Humans can perform complex internal reasoning before we speak: we disambiguate questions, recall facts, manipulate variables, and then verbalize only the polished answer. Intuitively, this unspoken internal thinking helps improve the accuracy and coherence of our spoken content. However, verbalizing these internal thoughts can make communication confusing and unnecessarily lengthy. Thus, it is desirable to enable SLMs a similar ability to think longer and silently before speaking out.

A naive way to do so is to teach SLMs to generate an *unspoken* chain-of-thought (CoT) reasoning (Wei et al., 2022) in text, and then generate the spoken responses based on the CoT reasoning. In this way, we can benefit from the CoT reasoning without making the spoken response overly long. While this does improve the speech output quality, as validated by our experiments in Section 5, the speech responses need to wait until the full text CoT is generated, creating an uncontrollable latency for the speech output.

To reduce and remove this latency, we propose STITCH: **S**imultaneous **Th**inking and **T**alking with **Ch**unked Reasoning. STITCH alternately generates unspoken reasoning chunks and spoken response chunks, allowing the spoken responses to condition on some thinking process. Since the audio duration of a chunk of spoken response is much longer than the time to generate the tokens in a chunk of spoken response, we use the remaining free time to generate the unspoken reasoning tokens. When a chunk of audio is played to the user, STITCH keeps generating the next unspoken reasoning chunk, achieving **simultaneous thinking and talking**. STITCH-S, a variant of STITCH, has the same latency as the models that cannot generate unspoken reasoning, while significantly outperforming those baselines on math reasoning datasets.

We summarize our contributions as follows:

1. We propose STITCH, an SLM generation pipeline that outputs unspoken reasoning chunks and spoken response chunks alternately. To the best of our knowledge, we are the first to introduce the unspoken thinking process to SLMs.

2. STITCH-S, while improving the generation with unspoken reasoning, has a latency the same as an SLM that does not generate text reasoning by design.

3. On five math QA datasets, STITCH consistently outperforms the baseline models that cannot reason with text CoT.

## 2 RELATED WORK: SPOKEN LANGUAGE MODELS

Our paper improves SLMs by generating text reasoning before generating the speech output. Current SLMs first generate some speech tokens, and the speech tokens will be synthesized into a waveform by the speech decoder. In this section, we recap how existing SLMs generate speech tokens. The mainstream SLMs nowadays adopt two types of speech generation methods: (1) **Interleaved** text-

speech tokens (Zeng et al., 2024; Li et al., 2025b) and the (2) **Thinker-Talker** architecture (Xu et al., 2025; Fang et al., 2025b).

In the **interleaved** decoding paradigm, the SLM backbone model generates a chunk of text tokens and a chunk of speech tokens alternately. The text tokens serve as guidance for future speech tokens by transcribing what the speech token will say. For example, GLM-4-Voice (Zeng et al., 2024) interleaves between generating $N_{\text{text}} = 13$ text tokens and $N_{\text{speech}} = 26$ speech tokens. After a chunk of speech tokens is generated, it is immediately synthesized into audio by the speech decoder and streamed to the user, enabling low latency and real-time interaction. A figurative illustration of this output format is shown in Figure 2(a). When concatenating the chunk of text tokens, they should correspond to the transcription of the speech tokens. The ratio of the text tokens and speech tokens is carefully selected such that the text tokens are always faster than the speech tokens to **ensure that the content of the speech tokens has already appeared in previous text tokens**. Once all the text tokens are generated, the model will continue to generate the remaining speech tokens.

The **thinker-talker** SLM uses a *thinker* model to take speech input and generate text tokens, and uses a *talker* model to convert the text tokens and/or their continuous representations into speech tokens (Xu et al., 2025; Fang et al., 2025b). The talker model functions like a Text-to-Speech (TTS) model that converts the textual output from the thinker into speech tokens, and the output by the thinker and talker will be highly aligned in their semantics, while there may be slight differences.

The text-speech interleaved SLMs and thinker-talker SLMs differ in their architecture, but they share a similarity: they generate text and then the speech, and the text is highly aligned with the speech. In this paper, we focus on enabling SLMs to generate an unspoken thinking process while speaking.

While reasoning, also sometimes referred to as thinking, is widely explored in text-only LLMs (OpenAI, 2024c; Wu et al., 2024; Guo et al., 2025; Hong et al., 2025), reasoning is only included in audio LLMs recently (Wen et al., 2025; Li et al., 2025a; Xie et al., 2025). Audio LLMs are multimodal LLMs that can take audio and speech as input, but they cannot generate speech outputs (Changli et al., 2023; Chu et al., 2024; Xu et al., 2025; Lu et al., 2025); this marks a significant difference from SLMs studied in our paper. Currently, no prior work introduces unspoken reasoning ability to SLMs.

## 3 METHOD: SIMULTANEOUS THINKING AND TALKING WITH CHUNKED REASONING

While current SLMs also include some textual inner monologue (Défossez et al., 2024) or thinking (Xu et al., 2025) (as in thinker-talker) before generating the speech tokens, these text tokens simply transcribe what the model will say next. In other words, they do not generate *unspoken* reasoning before speaking. In this paper, we aim to improve the SLMs by generating some unspoken text reasoning before generating the speech responses (including the text and speech tokens).

Our proposed concept can be applied to both the interleaved and thinker-talker SLMs, but **we adopt the interleaved SLMs** in this paper for the following three reasons: (1) The interleaved SLM only contains a single backbone model to generate the text and speech tokens, simplifying the model architecture. (2) Wu et al. (2025) show that the thinker-talker SLM requires careful tuning to make it properly generate well-aligned text and speech output, making thinker-talker harder to fine-tune. (3) Qwen-2.5-Omni, the most widely used thinker-talker SLM, does not publicly release the speech tokenizer, adding difficulties to fine-tune the talker model (Liu, 2025; Shin, 2025).

To unify the wording, we will use *reasoning tokens* to refer to the **unspoken** reasoning (in text), *text tokens* to refer to the original text tokens of the interleaved SLMs, and *speech tokens* to refer to the speech tokens. Text tokens and speech tokens are the output of interleaved SLMs from prior works (Zeng et al., 2024), and the reasoning tokens are introduced in this paper. Both text tokens and reasoning tokens are texts, while they differ in their function: reasoning tokens are for the unspoken thinking process, and text tokens are the transcriptions of the spoken output.

### 3.1 A TRIVIAL METHOD: THINK BEFORE SPEAKING IN SPEECH (TBS)

Before introducing STITCH, we start from a trivial baseline method that introduces the thinking process to SLMs: **T**hinking in text **B**efore **S**peaking in speech (TBS). Introducing TBS can help us

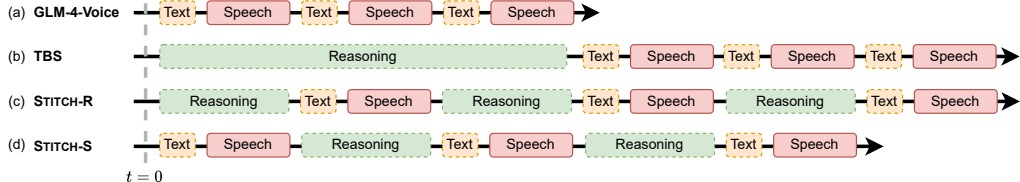

Figure 2: Different generation method explored in this paper. The arrow represents the timeline for the SLM to generate the tokens; this timeline should **not be confused** with the timeline that the end user receives the audio, i.e., the upper timeline in Figure 1. We plot tokens of the same type in a chunk using the same color. (a) GLM-4-Voice: Interleaving between text and speech token chunks (Section 2). This is the design of the original interleaved SLMs. (b) TBS: Generating a complete reasoning span and then interleaving between text and speech token chunks (Section 3.1). (c) STITCH-R: Alternating between reasoning token chunks, text token chunks, and speech token chunks (Section 3.2). (d) STITCH-S: Alternating between text token chunks, speech token chunks, and reasoning token chunks (Section 3.3).

understand the motivation and design of STITCH later. In TBS, given a user query in speech, the model first generates a complete trace of reasoning span. A complete reasoning span is enclosed by two special tokens [SOR] and [EOR], denoting the start and end of the reasoning span. The [EOR] token lets the model know that the reasoning has finished and it needs to start generating the tokens for the spoken response, including the text tokens and the speech tokens. The text tokens and the speech tokens are generated in an interleaved manner, following the original design of interleaved SLMs, with $N_{\text{text}}$ and $N_{\text{speech}}$ alternating with each other.

To teach SLMs to operate in TBS, we construct the training data $\mathcal{D}_{\text{TBS}}$ where each training instance has the form $(\mathbf{x}, \mathbf{z}, \mathbf{y})$, $\mathbf{x}$ is the speech token sequence of the user input, $\mathbf{z}$ is the reasoning token sequence, and $\mathbf{y} = [\mathbf{t}_1 \circ \mathbf{s}_1 \circ \mathbf{t}_2 \circ \mathbf{s}_2, \cdots]$ is the token sequence for the speech output that interleaves between $N_{\text{text}}$ text tokens ($\mathbf{t}_j$) and $N_{\text{speech}}$ tokens ($\mathbf{s}_j$) (The last text token chunk may be less than $N_{\text{text}}$ while the last speech token span can have more than $N_{\text{speech}}$ tokens); $\circ$ denotes concatenating two token sequences. We defer how we construct $\mathcal{D}_{\text{TBS}}$ from existing datasets until Section 4.1. A figurative illustration of the target output for TBS is in Figure 2(b).

During training, given an input $\mathbf{x}$, we train the SLM to predict the reasoning tokens $\mathbf{z}$ and then the interleaved text-speech tokens $\mathbf{y}$ using the standard cross-entropy loss.

### 3.2 STITCH-R: SIMULTANEOUS THINKING AND TALKING WITH REASONING FIRST

While inserting text reasoning before the speech response in TBS improves the quality of the speech response, the speech response can only be generated after the full reasoning has been generated. As later shown in Table 2, the reasoning by TBS can be long, creating significant latency before generating the speech response. To resolve the latency issue, we draw inspiration from the fact that humans can *think while speaking* and propose STITCH-R.

STITCH-R realizes this "*thinking when speaking*" by alternating fixed-length ($N_{\text{reason}}$) *partial* reasoning spans, fixed-length ($N_{\text{text}}$) text token spans, and fixed-length ($N_{\text{speech}}$) speech token spans. The partial reasoning spans are for inner thinking, while the text and speech token spans are for the spoken response. Stitching the partial reasoning spans together will form a complete CoT reasoning. A figurative illustration of the output of STITCH-R is shown in Figure 2(c), and some samples generated by STITCH-R are shown in Table 5 in the Appendix. The "*R*" in STITCH-R stands for "*reasoning first*" since it generates a partial reasoning chunk before speaking; this is used to distinguish the "*speaking first*" STITCH-S that will be introduced in Section 3.3.

We explain how simultaneous thinking and speaking is achieved by STITCH-R, which can be most easily understood with Figure 1. After the SLM receives the speech input, it first generates $N_{\text{reason}}$ reasoning tokens, $N_{\text{text}}$ text tokens, and $N_{\text{speech}}$ speech tokens for the first audio output chunk $S_1$. Those speech tokens will be passed through the speech decoder to synthesize the audio with duration $t_{\text{chunk}}$ seconds and played to the user. When synthesizing the speech waveform and playing the audio for $S_1$, the SLM continues to generate the next $N_{\text{reason}}$ reasoning tokens; **this is where simultaneous**

**thinking and speaking happens.** After generating the second reasoning chunk, the model continues to generate $N_{\text{text}}$ text tokens, and $N_{\text{speech}}$ speech tokens for the next audio chunk $S_2$. As long as the time to generate $N_{\text{reason}} + N_{\text{text}} + N_{\text{speech}}$ tokens, denoted as $t_{token}$, is less than $t_{\text{chunk}}$, the next audio chunk $S_2$ can be played seamlessly after the previous audio chunk $S_1$.

Let's understand with some concrete numbers. On an A100-80G GPU, our model generate around 80 tokens per second using vLLM (Kwon et al., 2023), and the duration of the audio for $N_{\text{speech}} = 26$ speech tokens is around $t_{\text{chunk}} = 2$ seconds, so we can generate around 160 tokens in $t_{\text{chunk}} = 2$ seconds. The number of text tokens and speech tokens sums to $N_{\text{text}} + N_{\text{speech}} = 13 + 26 = 39$, so we have time to generate at most $80 \times 2 - 39 = 121$ reasoning tokens.[1]

Each partial reasoning chunk is enclosed in two special tokens: [SOPR] (start of partial reasoning) and [EOPR] (end of partial reasoning). If the reasoning chunk is fully generated, an additional [EOR] is used to denote the end of the full reasoning CoT.

To construct the training data for STITCH-R, we simply split the full reasoning CoT $\mathbf{z}$ in $\mathcal{D}_{\text{TBS}}$ into chunks with $N_{\text{reason}}$ tokens $\{\mathbf{z}_1, \mathbf{z}_2, \cdots\}$, where each $\mathbf{z}_i$ except the last chunk has $N_{\text{reason}}$ tokens. Next, we interleave those chunks with the interleaved text-speech token sequence $\mathbf{y} = [\mathbf{t}_1 \circ \mathbf{s}_1 \circ \mathbf{t}_2 \circ \mathbf{s}_2, \cdots]$ to create the interleaved data of the form $[\mathbf{z}_1 \circ \mathbf{t}_1 \circ \mathbf{s}_1 \circ \mathbf{z}_2 \circ \mathbf{t}_2 \circ \mathbf{s}_2 \circ \cdots]$, as shown in Figure 2(c). If the number of the reasoning span is more than the number of text spans, this indicates that the reasoning token spans *think slower* than the text token spans, so we remove the sample from the training data.[2] The model is fine-tuned to auto-regressively predict the interleaved reasoning-text-speech token spans using standard language modeling cross-entropy loss.

### 3.3 STITCH-S: SIMULTANEOUS THINKING AND TALKING WITH SPEAKING FIRST

STITCH-R can have a latency shorter than that of TBS, since STITCH-R only needs to wait at most $N_{\text{reason}}$ tokens before generating the text and speech tokens. However, compared with the original text-speech interleaved SLMs that directly generate the text and speech tokens for the spoken response, STITCH-R still needs to wait for $N_{\text{reason}}$ reasoning tokens.

To fully remove the latency of waiting for the first partial reasoning span, we propose an alternative generative pipeline that directly starts to generate the text and speech token chunks and then generates the first reasoning chunk; the model continues to interleave this generation pattern. We call this STITCH-S since it generates a speech response first, and an illustrative figure is shown in Figure 2(d).

To construct the training data for STITCH-S, we similarly split the full reasoning $\mathbf{z}$ in $\mathcal{D}_{\text{TBS}}$ into chunks with $N_{\text{reason}}$ tokens, and insert a text token chunk of length $N_{\text{text}}$ and a speech token chunk of length $N_{\text{speech}}$ **before** each reasoning span. Following previous notation, the target output for STITCH-S looks like $[\mathbf{t}_1 \circ \mathbf{s}_1 \circ \mathbf{z}_1 \circ \mathbf{t}_2 \circ \mathbf{s}_2 \circ \mathbf{z}_2 \circ \mathbf{t}_3 \circ \mathbf{s}_3, \cdots]$. If a training instance's number of reasoning chunks is more than the number of text chunks, the above data construction pipeline will result in an instance that has a reasoning span after the text chunks are finished, which does not make sense since the thinking process hasn't ended while the content to be spoken is finished. We remove this kind of sample from the training dataset.

## 4 EXPERIMENT SETUP

### 4.1 TRAINING

We fine-tune a separate model for TBS, STITCH-R, and STITCH-S from GLM-4-Voice-9B (Zeng et al., 2024). During fine-tuning, we fully fine-tune the backbone transformer model (Vaswani et al., 2017) of GLM-4-Voice and freeze the speech encoder (Zeng et al., 2025) and speech decoder (Du et al., 2024; Kong et al., 2020). The detailed hyperparameters for fine-tuning are listed in Appendix C.

We include three types of tasks in the training data: general dialogue dataset (VoiceAssistant400K (Xiezhifei, 2024)), mathematical reasoning (Tulu-3-Persona Math, Tulu-3-Persona-Math-

---

[1] Since the token generation and audio synthesis can be run in parallel, we do not need to consider $t_{\text{syn}}$, the time to synthesize the audio, as long as $t_{\text{syn}} < t_{\text{chunk}}$.

[2] The number of text spans is always less than or equal to the number of speech spans by design. This can be done by properly selecting the ratio between $N_{\text{text}}$ and $N_{\text{speech}}$.

Grade, and Tulu-3-Persona-Algebra (Lambert et al., 2024)), and knowledge-intensive question answering (Natural Question (Kwiatkowski et al., 2019) and TriviaQA (Joshi et al., 2017)).

We introduce how to construct $\mathcal{D}_{\text{TBS}}$, the training data for TBS, while the training data for STITCH-R and STITCH-S can be constructed from $\mathcal{D}_{\text{TBS}}$ following the procedures in Section 3.2 and 3.3.

To train TBS, each sample in the dataset needs to have the spoken query $\mathbf{x}$, text reasoning $\mathbf{z}$, and the spoken responses $\mathbf{y}$. For VoiceAssistant400K, which is already a speech-to-speech dialogue dataset, we generate the text reasoning using GPT-4o. For mathematical reasoning datasets, which only contain a question and its long answer in text, we synthesize the question audio $\mathbf{x}$ using GPT-4o-mini-TTS (OpenAI, 2024a), use the ground truth long answer as the text CoT $\mathbf{z}$, and rewrite the long answer and synthesize its audio to form the spoken response $\mathbf{y}$. For knowledge-intensive QA datasets, which are also in text form, we synthesize the question audio $\mathbf{x}$ using GPT-4o-mini-TTS, generate the text CoT $\mathbf{z}$ with GPT-4o, and rewrite the answer and synthesize its audio to form the spoken response $\mathbf{y}$. All the prompts used for constructing the datasets are in Appendix B.1. The full training dataset contains around 400K instances.

## 4.2 EVALUATION

We include reasoning and non-reasoning datasets for evaluation. We use the former to evaluate whether reasoning before responding in speech can improve the spoken answer quality for reasoning tasks. We include five math reasoning datasets: AddSub (Hosseini et al., 2014), SinglEq (Koncel-Kedziorski et al., 2015), MultiArith (Roy & Roth, 2015), SVAMP (Patel et al., 2021), and GSM8K (Cobbe et al., 2021), which are commonly used to evaluate the math reasoning ability of LLMs (Wei et al., 2022; Kojima et al., 2022; Wang et al., 2023). The questions and answers in these datasets are not too difficult and can be described clearly in speech.Since these datasets are originally in textual form, we synthesize the question audio using GPT-4o-mini-TTS.

For non-reasoning datasets, we include three knowledge-intensive QA datasets (Llama Questions (Nachmani et al., 2024), TriviaQA (Joshi et al., 2017), and WebQuestions (Berant et al., 2013)) and AlpacaEval (Li et al., 2023), a dialogue dataset, from OpenAudioBench (Li et al., 2025b). We use these datasets to understand whether generating unspoken reasoning before responding in speech has any impact on non-reasoning datasets. The dataset statistics are in Appendix Table 4.

For all reasoning datasets and knowledge-intensive QA datasets, we evaluate the SLM's performance using the accuracy. To calculate the accuracy, we use GPT-4o to judge whether the SLM's prediction matches the ground truth answer. For AlpacaEval, we follow the setting in OpenAudioBench to use LLM-as-a-judge (Chiang & Lee, 2023; Liu et al., 2023; Zheng et al., 2023) to rate the response from 1 to 10, and report the GPT-4o-score scaled to 10 and 100, where higher is better.

When evaluating the responses from SLMs, we use the text tokens for evaluation. Alternatively, we can synthesize the audio from the speech tokens, transcribe the audio, and evaluate those transcriptions. However, since the SLM's text tokens are designed to align with the speech output, we only report the performance of the text tokens here. The general trend observed in the following experiment results holds when we evaluate the transcription, while the latter has a slight performance drop of around 3% to 6%, as shown in Table 7 in the Appendix.

## 4.3 COMPARED MODELS

We compare six models, which are based on the same base model: GLM-4-Voice, an interleaved SLM. GLM-4-Voice has an $N_{\text{text}} = 13$ and $N_{\text{speech}} = 26$. All the models we use follow the $N_{\text{text}}$ and $N_{\text{speech}}$ in GLM-4-Voice.

(1) **GLM-4-Voice**: The model takes speech input and generates interleaved text-speech tokens. This model cannot generate unspoken reasoning by design. (2) **No reasoning**: We fine-tune GLM-4-Voice on a dataset without reasoning, $\mathcal{D}_{\text{no-reason}}$. Each instance in $\mathcal{D}_{\text{no-reason}}$ has the form $(\mathbf{x}, \mathbf{y})$, which is constructed from $\mathcal{D}_{\text{TBS}}$ by removing the text reasoning $\mathbf{z}$. The model is trained to predict interleaved text-speech tokens $\mathbf{y}$ based on the input $\mathbf{x}$. (3) **TBS**: see Section 3.1. (4) **Mix reasoning**: We mix half of $\mathcal{D}_{\text{no-reason}}$ and half of $\mathcal{D}_{\text{TBS}}$ to fine-tune GLM-4-Voice. We use a system prompt to control whether the model should generate reasoning before generating the interleaved text-speech tokens. This allows us to control the model's behavior during inference.

(a) The accuracy on math QA.

| Id | Config | Latency | Use Reasoning | | Datasets | | | | | Avg. |
|----|--------|---------|---------------|------|---------|------------|---------|-------|-------|------|
| | | | Train | Infer. | AddSub | MultiArith | SinglEq | SVAMP | GSM8K | |
| colspan | | | | | *Baselines* | | | | | |
| 1 | GLM-4-Voice | $N_{\text{text}} + N_{\text{speech}}$ | – | ✗ | 59.42 | 62.00 | 71.00 | 44.00 | 29.00 | 53.08 |
| 2 | No reasoning | $N_{\text{text}} + N_{\text{speech}}$ | ✗ | ✗ | 66.06 | 70.69 | 77.98 | 64.43 | 35.73 | 62.98 |
| 3 | Mix reasoning | $N_{\text{text}} + N_{\text{speech}}$ | ✔+✗ | ✗ | 69.72 | 79.31 | 87.16 | 63.55 | 37.33 | 67.41 |
| | | | | | *Full Reasoning before Speaking* | | | | | |
| 4 | TBS | $N_{\text{full}} + N_{\text{text}} + N_{\text{speech}}$ | ✔ | ✔ | 79.82 | 85.63 | 89.91 | 75.29 | 64.94 | 79.12 |
| 5 | Mix reasoning | $N_{\text{full}} + N_{\text{text}} + N_{\text{speech}}$ | ✔+✗ | ✔ | 76.15 | 89.08 | 88.99 | 70.90 | 62.23 | 77.47 |
| | | | | | *Proposed Methods: Simultaneous Thinking and Speaking* | | | | | |
| 6 | STITCH-R | $N_{\text{reason}} + N_{\text{text}} + N_{\text{speech}}$ | ✔ | ✔ | 78.90 | 88.51 | 93.58 | 73.83 | 58.70 | 78.70 |
| 7 | STITCH-S | $N_{\text{text}} + N_{\text{speech}}$ | ✔ | ✔ | 81.65 | 87.93 | 91.74 | 72.15 | 56.72 | 78.04 |

(b) The performance on non-reasoning tasks. For factual QA, including Llama Question, TriviaQA, and WebQuestion, we report the accuracy. For AlpacaEval, we report the GPT-4o score.

| Id. | Config | Use Reasoning | | Datasets | | | | Avg. |
|-----|--------|---------------|------|----------------|----------|-------------|------------|------|
| | | Train | Infer. | Llama Question | TriviaQA | WebQuestion | AlpacaEval | |
| | | | | *Inference without Reasoning* | | | | |
| 1 | GLM-4-Voice | – | ✗ | 74.25 | 47.10 | 50.95 | 48.59 | 55.22 |
| 2 | No reasoning | ✗ | ✗ | 73.33 | 46.00 | 47.80 | 53.62 | 55.19 |
| 3 | Mix reasoning | ✔+✗ | ✗ | 73.33 | 47.80 | 48.00 | 54.77 | 55.98 |
| | | | | *Full Reasoning before Speaking* | | | | |
| 4 | TBS | ✔ | ✔ | 74.33 | 51.50 | 52.15 | 56.33 | 58.58 |
| 5 | Mix reasoning | ✔+✗ | ✔ | 69.67 | 48.80 | 50.75 | 56.03 | 56.31 |
| | | | | *Proposed Methods: Simultaneous Thinking and Speaking* | | | | |
| 6 | STITCH-R | ✔ | ✔ | 70.00 | 49.60 | 50.25 | 54.02 | 55.97 |
| 7 | STITCH-S | ✔ | ✔ | 73.33 | 50.00 | 50.15 | 54.62 | 57.03 |

Table 1: Performance on reasoning dataset (1a) and non-reasoning datasets (1b). The *latency* column reports how many tokens the SLM should generate until the first chunk of speech can be synthesized, where $N_{\text{full}}$ denotes the uncontrollable length of the full reasoning. The *"use reasoning"* columns denote whether text reasoning $\mathbf{z}$ is in the training data or is generated during inference.

By comparing the performance of this model when inference with and without reasoning, we can understand if unspoken reasoning really helps SLM's speech output. (5) **STITCH-R**: see Section 3.2. (6) **STITCH-S**: see Section 3.3.

During training, we set $N_{\text{reason}} = 100$ for STITCH-R and STITCH-S. In Section 3.2, we have derived that the length of the reasoning token can be at most 121 tokens when running on an A100-80G GPU, so we select $N_{\text{reason}} = 100$. We do not compare with other SLMs as they use different training data and have distinct model architectures, making the comparison with our models less meaningful.

## 5 MAIN RESULTS

We present the main results in Table 1. We summarize several key observations as follows:

**TBS significantly outperforms the baselines without reasoning on math QA.** In row 4, TBS achieves an average accuracy of 79.12%, outperforming GLM-4-Voice (row 1) by 26.04% and the *"no reasoning"* model (row 2) by 16.14%. Most significantly, the accuracy of TBS almost doubles compared to the two baselines without reasoning on GSM8K. Using McNemar's test, we found that the TBS outperforms GLM-4-Voice and the no-reasoning baselines in a statistically significant way, with $p$-value less than 0.05. This shows that reasoning before speaking can help the SLM respond better to questions that require reasoning.

| Datasets | Token Count | | | | | | Quality Assesment | | | |
|---|---|---|---|---|---|---|---|---|---|---|
| | MultiArith | | GSM8K | | TriviaQA | | MultiArith | GSM8K | MultiArith | GSM8K |
| Metrics | Reason | Trans | Reason | Trans | Reason | Trans | UTMOSv2 | | GPT-4o-score | |
| GLM-4-Voice | – | 66.17 | – | 92.38 | – | 51.67 | 3.10 | 3.05 | 4.60 | 4.28 |
| No reasoning | – | 45.67 | – | 60.22 | – | 28.64 | 3.15 | 3.13 | 4.91 | 4.49 |
| TBS | 178.95 | 45.09 | 360.04 | 70.49 | 68.03 | 27.55 | 3.14 | 3.16 | 4.90 | 4.78 |
| STITCH-R | 157.32 | 49.77 | 322.40 | 74.36 | 67.56 | 27.65 | 3.10 | 3.12 | 4.94 | 4.74 |
| STITCH-S | 203.28 | 80.87 | 334.71 | 61.85 | 69.45 | 28.21 | 3.17 | 3.11 | 4.88 | 4.66 |

Table 2: Left: The average number of total reasoning tokens (summed over partial reasoning spans for STITCH-R and STITCH-S) and the text tokens. Right: The quality scores of the speech responses.

**Reasoning before speaking is better than no reasoning before speaking, even if the training data is the same.** The strong result of TBS, compared with the "*no reasoning*" baseline, may raise a question on whether TBS is winning because the training data includes additional reasoning **z**, which are not included in the training data for the "*no reasoning*" baseline. This question can be answered by comparing the results of the "*mix reasoning*" model with and without text reasoning before generating the speech response (row 3 vs. row 5). Since the two different inference modes use the same *mix reasoning* model, the performance difference can be attributed to the reasoning before the spoken response during inference. On average, reasoning before speaking (row 5) obtains an accuracy of 77.47%, outperforming not using reasoning by more than 10% (row 3). For the most difficult GSM8K, the improvement of using reasoning is 24.9%. The results again confirm the importance of generating reasoning before the speech response for reasoning tasks.

**STITCH-R (row 6) shows almost no performance drop on average.** STITCH-R alternates between partial text reasoning and text-speech tokens. While the reasoning is split by text-speech tokens, the average performance only drops by 0.42% compared to TBS. In Table 2, we report the average number of reasoning tokens on three testing datasets. Both TBS and STITCH-R generate 150 to 300 reasoning tokens for math reasoning datasets on average, but STITCH-R only needs to wait for the first $N_{token} = 100$ reasoning tokens before starting to generate the first chunk of the speech output, yielding a smaller latency for STITCH-R. Importantly, this speedup comes with almost no loss in accuracy. Still, on a more difficult dataset (GSM8K), the performance drop is larger, but it still significantly outperforms the non-reasoning baselines (row 1 and row 2). Again, using McNemar's test, we find that the STITCH-R outperforms the no-reasoning baselines in a statistically significant way. We discuss two samples from STITCH-R in Section 6.

**STITCH-S shows almost no performance drop on average without adding any inference latency.** Remarkably, STITCH-S shows an average accuracy of 78.04% (row 7), slightly lower than TBS by 1.08% (row 4), but is still much better than the "*no reasoning*" baseline by 15.06% (row 2). We see a larger performance drop in GSM8K around 8% compared with TBS, but again, still significantly outperforms *no reasoning* baseline by 19.99%, again in a statistically significant way based on McNemar's test. Recall that the first packet latency of STITCH-S is exactly the same as the baselines that do not generate text reasoning; both of them only need to wait for $N_{\text{text}} + N_{\text{speech}}$ tokens before synthesizing the speech waveform. This exciting result indicates that we can improve the answer quality **without sacrificing any inference latency**. By scrutinizing the responses from STITCH-S, we find that the first text token chunk is often a rephrase of the question, which can be generated without any reasoning. This explains why STITCH-S does not show a severe accuracy drop compared with STITCH-R.

**On non-reasoning datasets, reasoning models perform comparably well or slightly outperform non-reasoning baselines.** Table 1b shows the performance on non-reasoning datasets. We find that TBS (row 4), STITCH-R (row 6), and STITCH-S (row 7) all have performance comparable to or better than the GLM-4-Voice (row 1) and the *no reasoning* baseline (row 2). This shows that fine-tuning SLMs to think in text before responding does not harm the performance on non-reasoning datasets.

**Generating text reasoning before generating the speech responses does not harm the perceptual quality and the fluency of the speech response.** In the rightmost block in Table 2, we report some quality assessment of the responses generated by the SLMs. We report (1) the perceptual quality

of the speech response evaluated by UTMOSv2 (Baba et al., 2024), an automatic MOS evaluation model, and (2) the fluency score of the text tokens judged by GPT-4o (Chiang & Lee, 2023). The fluency score focuses on whether the text tokens form a fluent narrative. Both metrics range from 1 to 5, with 5 being the best. We observe that all fine-tuned models have a similar UTMOSv2 score over 3, similar to the original GLM-4-Voice, indicating that our fine-tuning does not harm the perceptual quality of the speech output. We also observe that the fluency scores for STITCH-R and STITCH-S are high, indicating that alternating between the text reasoning token spans and the text-speech token spans does not harm the fluency of the text tokens.

**Human evaluation shows that STITCH-S is more responsive than STITCH-R, followed by TBS.** We conduct a human evaluation on the responsiveness of different models. We present the human evaluator with a question from the user and two audio responses from different models, asking them to compare the responsiveness of the two models on a seven-point scale, where +3 indicates that model A is more responsive than model B and -3 indicates that model B is more responsive than model A. To simulate the first packet latency that the user may experience, each audio includes a silence in the beginning, accounting for the prefilling and decoding time of the SLM and the speech synthesis latency. We compare the following pairs: TBS / STITCH-R, TBS / STITCH-S, STITCH-R / STITCH-S, and STITCH / no-reasoning. For each pair, we use the same 100 questions from GSM8K, and we recruit 3 human evaluators to judge the responsiveness of a pair of models. The results are presented in Table 6 in the Appendix, and we summarize the results as follows: 1. STICH-R and STITCH-S are both more responsive than TBS. 2. STITCH-S is more responsive than STITCH-R. 3. STITCH-S and no-reasoning are equally responsive.

**The reasoning segments finish much earlier than the text segments.** Based on the statistics in Table 2, we can see that the average number of reasoning chunks are less than the number of text chunks. For example, on GSM8K, STITCH-R has on average $322.40/100 = 3.22$ reasoning chunks, while the average number of text chunks is $74.36/13 = 5.72$. During inference time, we do not find the interleaved text-speech chunks to end before the reasoning chunks finish, i.e., before the EOR is emitted. This is mainly because when we construct the training data for STITCH-R and STITCH, we remove the samples whose reasoning chunks are more than the text chunks, i.e., those samples that have reasoning slower than the response. As a result, after training, the model will learn that it must finish the reasoning before concluding its spoken response.

## 6 QUALITATIVE RESULTS

In this section, we provide two outputs from our STITCH-R in Table 5 and a sample for STITCH-S in Table 8 in the Appendix. The top sample in Table 5 shows a typical pattern of the output from STITCH-R, where all the calculation results from the text token chunks are already calculated in previous reasoning chunks. When generating the text tokens, the model simply needs to rewrite the lengthy reasoning into more concise sentences that are easier for spoken communication.

The bottom sample in Table 5 shows a somewhat rare pattern, where the second text token chunk includes a new calculation result (the number 120) that has not been explicitly calculated in previous reasoning chunks. This shows that, besides thinking during the reasoning chunks, the model sometimes also incorporates the text tokens for thinking. However, this pattern is rare: We use GPT-4o to parse the 1310 responses to GSM8K and find that only 5% of the samples illustrate this pattern. The same pattern is also rare on STITCH-S.

Last, we find that the first text token chunk tends to rephrase the information of the question without adding new calculation results, which can be seen in the two samples in Table 5. This shows that the first text token chunk can be generated without relying on additional reasoning. This can explain why STITCH-S can have a performance similar to TBS and STITCH-R: since the first text token chunk can be generated without thinking, we can directly generate those texts without conditioning on any reasoning.

## 7 ADJUSTING THE LENGTH OF THE REASONING CHUNK DURING INFERENCE

During training, we fix $N_{token}$, the number of tokens in each partial reasoning chunk, to 100 tokens. In this section, we ask whether we can adjust $N'_{token}$ at test time without retraining the model.

One might want to adjust $N'_{token}$ at test time if the hardware is slower and the time to generate $N_{token} + N_{\text{text}} + N_{\text{speech}}$ tokens may exceed the duration $t_{\text{chunk}}$ of the audio for $N_{\text{speech}}$ tokens. In this case, the model deployer may want to reduce $N'_{token}$ during inference time to ensure that generating the $N'_{token} + N_{\text{text}} + N_{\text{speech}}$ does not exceed the duration $t_{\text{chunk}}$. Thus, it would be interesting to see if we can train a single model with a fixed $N_{token}$ and vary $N'_{token}$ during inference.

Recall that each reasoning chunk ends with [EOPR]. To adjust the length of the reasoning chunk to $N'_{token} \leq N_{token}$ during the inference time, we can append [EOPR] after $N'_{token}$ reasoning tokens are generated. Once the model sees [EOPR], it will cease generating the reasoning tokens and switch to generating text-speech interleaved tokens. We intervene in the model's generation by the above process whenever the model generates a reasoning chunk until the [EOR] token is generated.

We vary the number of $N'_{token}$ from 60 to 100 with a 10 increment on the math reasoning datasets for STITCH-R and STITCH-S. The results are shown in Appendix Figure 3a and 3b; we summarize the main findings below. We see that the accuracy on all tasks mostly rises when increasing $N'_{token}$ from 60 to 100. While the performance of $N'_{token} < 100$ is mostly worse than the performance of $N_{token} = 100$, using $N'_{token} < 100$ is still better than the "*no reasoning*" baseline when $N'_{token} \geq 70$. We also observe that when $N'_{token} \geq 80$ tokens, the accuracy recovers to 90% of the accuracy of $N_{token} = 100$. The results indicate that the model deployer can control the number of $N'_{token}$ in a reasonable range during inference without retraining the model with some accuracy trade-off.

## 8 USING REASONING FROM OTHER MODELS

Previously, we showed that STITCH-R and STITCH-S can generate reasoning before responding in speech. Here, we discuss whether we can improve the speech response by using the reasoning from *other* models. The motivation is that sometimes the reasoning generated by the SLM itself is not good enough or takes too long to generate, and we may use other stronger or smaller models to generate the reasoning. We call the model used to generate the reasoning the *"reasoning augmentation model"*.

We discuss how to improve STITCH-R by using a reasoning augmentation model. Given a user input speech, we first use an ASR model to transcribe the audio and feed the transcription into the reasoning augmentation model to generate a response, which will be used as the reasoning for STITCH-R. Next, we crop the reasoning into chunks of $N'_{token}$ enclosed by [SOPR] and [EOPR], and then feed the speech input and the first reasoning chunk to STITCH-R and generate the first text-speech interleaved token span. After the first text-speech interleaved tokens are generated, we append the second reasoning chunk to the input and generate the second text-speech interleaved token spans. We continue the above process to generate the full response.

In the following experiment, we replace the text reasoning of STITCH-R using four models: GLM-4-9B-Chat (GLM et al., 2024), Llama-3.1-8B, Llama-3.2-3B, and Llama-3.2-1B (Grattafiori et al., 2024). The results are shown in Figure 3c in the Appendix, and we summarize the main findings below. In general, stronger reasoning augmentation models (GLM-4-9B-Chat and larger Llama models) can improve the accuracy of the spoken response, while weaker reasoning augmentation model like Llama-3.2-1B worsens the performance of STITCH-R. This shows that STITCH-R can incorporate the reasoning ability of stronger models to improve the spoken answer. The experiment also shows that STITCH-R really considers the contents in the reasoning to form the spoken response, instead of independently generating the reasoning and not using that reasoning in the spoken responses.

## 9 CONCLUSION

In this paper, we introduce STITCH, a generation framework for SLM that enables simultaneous thinking and speaking. We achieve simultaneous thinking and speaking by using the long duration of an audio chunk to generate additional reasoning tokens before generating the speech tokens for the next audio chunk. By design, STITCH-S induces no additional latency compared to the models that cannot generate reasoning before they speak. On five math reasoning QA datasets, we show that STITCH-R and STITCH-S significantly improve the accuracy over non-reasoning baselines by over 15%. On non-reasoning datasets, STITCH-R and STITCH-S perform comparably well with non-reasoning baselines. Overall, the low latency and high performance of STITCH show a promising future direction for spoken language models.

## 10 ETHICS STATEMENT

Our paper fully follows the ICLR Code of Ethics. We do not see any potential harm in the paper.

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

## A  CHANGE LOG

The following change log is included to help readers understand the modifications made throughout the author-reviewer discussion period.

**Sep. 24th**    The initial submission version.

**Nov. 19th**    First revision based on the initial reviews from the reviewer. The modifications are highlighted in blue. The main differences are summarized as follows:

1. **Explain the difficulty of fine-tuning Qwen-2.5-Omni**: In Section 3, we add a sentence to explain why we do not fine-tune Qwen-2.5-Omni.

2. **Statistical tests in Section 5**: We run McNemar's test to see if the performance between no-reasoning baselines and other models is statistically significant and report the results in Section 5.

3. **Explaining that reasoning always finishes before text-speech segments end**: In Section 5, we include a paragraph explaining that the reasoning segments generated by STITCH will always be less than the text/speech segments.

4. **Human evaluation in Section 5**: We add a paragraph in Section 5, where we run a human evaluation comparing the responsiveness of different pairs of models.

5. **Explain when to use STITCH-R/STITCH-S**: Based on all the previous experiment results, in the last paragraph in Section 5, we explain when to use STITCH-R and STITCH-S.

6. **Qualitative results moved to Section 6**: We move the qualitative results into the main content, where we illustrate how the reasoning of STITCH-R and STITCH-S precedes the text responses most of the time.

7. **Add Figure 5**: We add Figure 5 to show that different $t_{\text{syn}}$ will not break the thinking while speaking pattern of STITCH as long as $t_{\text{syn}} < t_{\text{chunk}}$.

**Feb. 8th**    Camera-ready version. Minor edits for typos and add links for datasets and demos. We remove the summary in Section 5 due to space constraints. The exact contents are as follows: *When choosing between STITCH-R and STITCH-S, if latency is the main consideration, choose STITCH-R; if a small latency is acceptable while the task accuracy is more important, choose STITCH-R.*

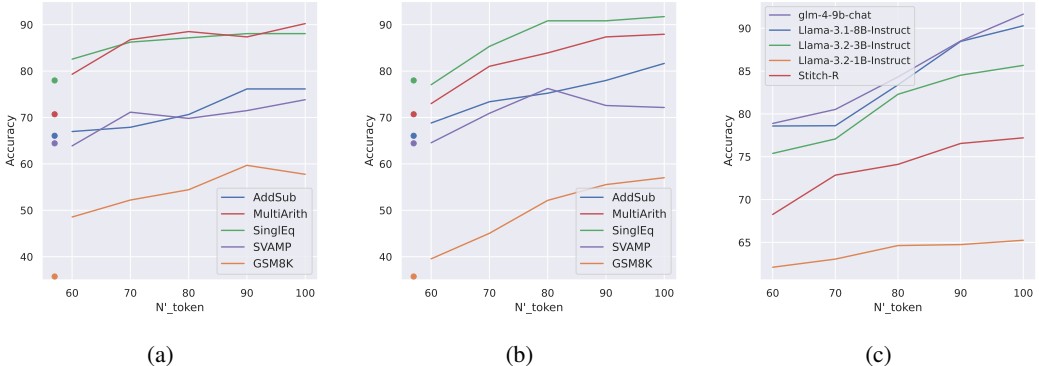

Figure 3: Figure 3a and 3b show the accuracy when varying $N'_{token}$ for STITCH-R and STITCH-S, respectively, and the dots in the figure are the performance of the "*no reasoning*" baseline (Section 7). Figure 3c shows the performance when using a reasoning augmentation model to generate the text reasoning for STITCH-R (Section 8); the accuracy is averaged over five math reasoning datasets.

## B DATASET

### B.1 TRAINING DATASET

We use three types of training data: dialogue datasets, math reasoning datasets, and knowledge-intensive QA datasets. The sizes of the datasets are in Table 3. We explain how we construct our training datasets from those existing datasets as follows.

**Dialogue Datasets** VoiceAssistant400K (Xiezhifei, 2024) is a spoken dialogue dataset built from a text dialogue dataset, including OpenAssistant (Köpf et al., 2023) and HH-RLHF (Bai et al., 2022). We subsample the dataset by filtering out (1) the instances whose `round` attribute is None or 0, (2) the instances from the subset `identity`, and (3) the instances whose response audio is less than 2 seconds. Given a user query and the response or answer from the dataset, we use GPT-4o to generate a text reasoning given some in-context samples. GPT-4o is instructed to generate CoTs that understand the user's intention, recall relevant information, and provide the final answer. The prompt we use is shown in Table 13.

**Math Reasoning Datasets** We include three math reasoning datasets for fine-tuning: Tulu-3-Persona Math, Tulu-3-Persona-Math-Grade, and Tulu-3-Persona-Algebra. The datasets include the question and the CoT answer in text. We directly use the ground truth CoT answer as the text reasoning. To obtain the spoken answer, we prompt GPT-4o to rewrite the answer such that the answer is suitable for speech communication and then synthesize it with GPT-4o-mini-TTS. The voice is sampled from the following list: *alloy, ash, ballad, coral, echo*. The prompt used to rewrite the spoken answer is shown in Table 10. The question audio is also synthesized with GPT-4o-mini-TTS, while the list of voices we sampled from is the same as the previous. The voice of the spoken question and response is selected randomly and independently. The total size of the math reasoning datasets used for fine-tuning is 220K.

**Knowledge-Intensive Datasets** We include two knowledge-intensive QA datasets, Natural Question (NQ) (Kwiatkowski et al., 2019) and TriviaQA (TQA) (Joshi et al., 2017), to preserve the SLM's ability to respond to knowledge-seeking questions. These two datasets are text-based datasets. To obtain the dataset for fine-tuning, we rewrite the ground truth answer from the dataset to make it more suitable for spoken communication, and then synthesize the question and answer audio using GPT-4o-mini-TTS. We sample from the same subset of voices randomly. The text reasoning is also generated by GPT-4o by prompting it with the question and ground truth answer, and asking it to generate an inner monologue that understands the user's intention, recalls relevant information, and provides the final answer. The prompts used for rewriting the answer and generating the reasoning are shown in Table 12 and Table 11, respectively. When subsampling from NQ and TQA, we simply

take the first 50K instances from NQ and the first 20K instances from TQA without shuffling the dataset before subsampling. The size of knowledge-intensive datasets is around 70K instances.

| Dataset | Size in Training Data | Download Link |
|---|---|---|
| VoiceAssistant | 177K | gpt-omni/VoiceAssistant-400K |
| Tulu3 Math | 150K | allenai/tulu-3-sft-personas-math |
| Tulu3 Algebra | 20K | allenai/tulu-3-sft-personas-algebra |
| Tulu3 GSM | 50K | allenai/tulu-3-sft-personas-math-grade |
| Natural Question | 50K | google-research-datasets/natural_questions |
| TriviaQA | 20K | mandarjoshi/trivia_qa |

Table 3: Training data sizes for each dataset. The download link in the above table is the link where we download the original datasets and then apply our filtering and preprocessing. Directly downloading from those links will not get the training dataset we use. Please follow the steps detailed in Appendix B.1 to construct the training data.

### B.2 TESTING DATASET

The testing dataset statistics are shown in Table 4. We release the audio of the math datasets we use at `https://huggingface.co/datasets/dcml0714/speech_math`. We use Kimi-Audio-Evalkit (Ding et al., 2025) to download OpenAudioBench. All our evaluation is run using Kimi-Audio-Evalkit. When evaluating the accuracy for math reasoning datasets and knowledge-intensive QA datasets, we use GPT-4o to judge if the model prediction matches the ground truth; this is supported by Kimi-Audio-Evalkit. The numbers of our result for GLM-4-Voice slightly differ from the numbers reported in Ding et al. (2025) since we use GPT-4o as the judge, while Ding et al. (2025) use GPT-4o-mini as the judge. In the original paper that proposes OpenAudioBench, Li et al. (2025b) also use GPT-4o as the judge, and our numbers for GLM-4-Voice are very close to those reported in Li et al. (2025b).

| Math Reasoning Datasets | | | | |
|---|---|---|---|---|
| AddSub | MultiArith | SinglEq | SVAMP | GSM8K |
| 109 | 174 | 109 | 299 | 1310 |

(a)

| OpenAudioBench | | | |
|---|---|---|---|
| Llama Questions | TriviaQA | Web Questions | AlpacaEval |
| 300 | 1000 | 1000 | 199 |

(b)

Table 4: Counts for Math Reasoning benchmarks (top) and OpenAudioBench (bottom).

## C TRAINING DETAILS

We use LlamaFactory (Zheng et al., 2024) to fine-tune the model. Fine-tuning a model using LlamaFactory (Zheng et al., 2024) takes 17 hours on 32 A100-80GB GPUs. To train with LlamaFactory, we use the speech tokenizer to encode the audio of the question and answer into speech tokens. Since LlamaFactory does not support fine-tuning GLM-4-Voice, the reader will need to slightly modify some source code in LlamaFactory to allow it to support GLM-4-Voice. We provide the exact `yaml` configuration file for fine-tuning below, including the detailed list of hyperparameters we use.

```
### model
model_name_or_path: THUDM/glm-4-voice-9b
trust_remote_code: true

```

**Question**: Josh decides to try flipping a house. He buys a house for $80,000 and then puts in $50,000 in repairs. This increased the value of the house by 150%. How much profit did he make?

**Response**:

[SOPR] ## Inner Monologue    This is a math question. I need to think step by step. To solve this problem, we need to determine the final value of the house after the repairs and then calculate the profit Josh made.
1. **Initial Purchase Cost**: Josh buys the house for $80,000.
2. **Cost of Repairs**: He spends $50,000 on repairs.
3. **Total Investment**: The total investment is the sum of the purchase cost and the repair cost [EOPR]

Josh spent $80,000 on the house and $50,    <audio_tokens> (...)

[SOPR] Total Investment = $80,000 + $50,000 = $130,000
4. **Increase in Value**: The value of the house increases by 150% after the repairs. To find the new value of the house, we calculate 150% of the initial purchase cost and add it to the initial purchase cost.    Increase in Value = 1.5 × $80, [EOPR]

000 on repairs, totaling $130,000. After repairs,    <audio_tokens> (...)

[SOPR]    000 = $120,000. New Value of the House = $80,000 + $120,000 = $200,000
5. **Profit Calculation**: The profit is the difference between the new value of the house and the total investment.    Profit = $200,000 - $130,000 = $70,000
Thus, Josh made a profit of [EOPR]

the house's value increased by 150%, making it $200    <audio_tokens> (...)

[SOPR]    $70,000.    #### $70,000 [EOPR][EOR]

,000. So, his profit is $200,000 minus    <audio_tokens> (...)    $130,000, which equals $70,000. The    <audio_tokens> (...)

answer is 70,000.    <audio_tokens> (...)

**Question**: Hannah's city is having a big display of fireworks for the 4th of July. They're going to set off 15 boxes of 20 fireworks each. Hannah's house is at the right angle to see 40% of the city's fireworks. Hannah will also set off 3 boxes of 5 fireworks each in her backyard. How many fireworks will Hannah see in total?

**Response**:

[SOPR]    ## Inner Monologue This is a math question. I need to think step by step. To solve this problem, we need to calculate the total number of fireworks Hannah will see from both the city's display and her own backyard display.
**Step 1: Calculate the total number of fireworks from the city's display.**
The city's display consists of 15 boxes of 20 fireworks each. To find the total number of fireworks from the city's display, we multiply the number of [EOPR]

Hannah's house will see 40% of the city's    <audio_tokens> (...)

[SOPR] boxes by the number of fireworks in each box:    15 boxes × 20 fireworks per box = 300 fireworks
**Step 2: Calculate the total number of fireworks from Hannah's backyard display.**
Hannah sets off 3 boxes of 5 fireworks each in her backyard. To find the total number of fireworks from her backyard display, we multiply the number of boxes by the number of fireworks in each box: [EOPR]

fireworks, which is 120. Plus, she has 3    <audio_tokens> (...)

[SOPR]    3 boxes × 5 fireworks per box = 15 fireworks
**Step 3: Calculate the total number of fireworks Hannah will see.**
Hannah will see 40% of the city's fireworks. To find out how many fireworks this is, we calculate 40% of the total fireworks from the city's display:    40% of 300 fireworks = [EOPR]

boxes of her own fireworks, adding 15 more. So in    <audio_tokens> (...)

[SOPR]    0.40 × 300 = 120 fireworks
**Step 4: Add the fireworks from the city's display and Hannah's backyard display.** Finally, we add the fireworks from the city's display and Hannah's backyard display to find the total number of fireworks Hannah will see: 120 fireworks from city display + 15 fireworks from backyard display = 135 fireworks    #### 135 [EOPR] [EOR]

total, she'll see 135 fireworks.    <audio_tokens> (...)

Table 5: Example outputs from STITCH-R. The text reasoning tokens are highlighted in green ; the text tokens are highlighted in orange ; the audio tokens, which are omitted for simplicity, are highlighted in red . We remove and insert some newlines for formatting.

```
5  ### method
6  stage: sft
7  do_train: true
8  finetuning_type: full
9
10 ### dataset
11 template: glm4voice
12 cutoff_len: 8192
13 overwrite_cache: true
14 preprocessing_num_workers: 16
```

```
15  dataloader_num_workers: 4
16  cache_dir: ./
17
18  ### output
19  logging_steps: 10
20  save_steps: 500
21  plot_loss: true
22  overwrite_output_dir: true
23  save_only_model: false
24  report_to: none
25
26  ### train
27  per_device_train_batch_size: 2
28  gradient_accumulation_steps: 8
29  learning_rate: 1.0e-5
30  num_train_epochs: 2.0
31  lr_scheduler_type: cosine
32  warmup_ratio: 0.1
33  bf16: true
34  ddp_timeout: 180000000
35  enable_liger_kernel: true
36  deepspeed: examples/deepspeed/ds_z2_config.json
```

## D  HUMAN EVALUATION

This section elaborates on how we conduct human evaluations to evaluate the responsiveness of different model pairs. We conduct pairwise evaluation of the responsiveness of the following model pairs: (1) STITCH-R vs TBS, (2) STITCH-S vs TBS, (3) STITCH-R vs STITCH-S, and (4) STITCH-S vs no-reasoning. We select the above model pairs because we believe they can form a partial order of the responsiveness of the four models.

We generate the responses of each model from the same 100 questions from GSM8K; the reason for selecting GSM8K is that the number of reasoning tokens required for this dataset is longer, as shown in Table 2, which may make the difference of responsiveness among the models more observable. For each audio, we insert a silence to represent the first packet latency. The first packet latency considers the prefilling and decoding time for the SLM, and the speech synthesis latency of the speech decoder. The prefilling and decoding latency is profiled using an A100 GPU with vLLM, while the speech synthesis latency is calculated based on the reported latency from Cosyvoice.

The human evaluators are given a question in GSM8K and two audios from a pair of models, and they are asked to judge the responsiveness of the two models on a 7-point comparison category rating, following the P.808 (Rec, 2018), using the following rubrics:

+3 : A is more responsive than B

+2 : A is somewhat responsive than B

+1 : A is slightly responsive than B

 0 : A and B are equally responsive

-1 : B is slightly responsive than A

-2 : B is somewhat responsive than A

-3 : B is more responsive than A

We use Amazon MTurk (AMT) to recruit the human evaluators, and each pair of comparisons is rated by three annotators. We require the annotators to have a HIT approval rate $\geq 98$ and to hold a master qualification on AMT. We also only include participants from the US, UK, Australia, and New Zealand in this study. The interface for the human evaluation is shown in Figure 4. Human evaluators are paid $1.70 for rating 8 pairs of responses, and we estimate that the hourly rate is roughly $12.75.

The results are summarized in Table 6. STITCH-S reaches a score of 1.687 against TBS, while STITCH-R obtains 1.164 compared to TBS, indicating that while both STITCH models are more

| Model A | STITCH-R | STITCH-S | STITCH-R | STITCH-S |
| Model B | vs
TBS | vs
TBS | vs
STITCH-S | vs
no-reasoning |
| --- | --- | --- | --- | --- |
| Mean score | 1.164 | 1.687 | -0.850 | 0.290 |

Table 6: The mean responsiveness scores of the four pairs of models. A positive score indicates that the upper model (model A) is more responsive than the lower one (model B), while a negative number means the lower model is more responsive. The larger the absolute value of the score is, the more obvious the responsiveness of the two models is for human evaluators.

responsive than TBS, the responsiveness of STITCH-S is slightly better than STITCH-R. The pairwise comparison between STITCH-R and STITCH-S also shows that STITCH-S is perceived as more responsive than STITCH-R. Comparing STITCH-S and the no-reasoning baseline, the average score is 0.290, while the 95% confidence interval is $[0.131, 0.449]$, meaning that the difference in the responsiveness is not very significant.

## E    DETAILED LATENCY CALCULATION

In the main content, we assume that the duration of each audio chunk $t_{\text{chunk}}$ synthesized from $N_{\text{speech}}$ tokens is the same, and we use $t_{\text{chunk}} = 2$ seconds in the paper. This is a simplification to make the discussion easier. In fact, for the first chunk of 26 speech tokens, the corresponding audio is about 1.6 seconds, while the later audio chunks all have a duration of about 2 seconds. As a result, a safer $N_{\text{reason}}$ on an A100-80G GPU, with token per second around 86, should be $86 \times 1.6 - (13 + 26) = 98.6$, which is still around 100, the number we use in our paper. It is also possible to make only the first reasoning chunk shorter while all the later reasoning chunks have $N_{\text{reason}} = 100$. If the model is deployed on H100-80G GPU, the token per second can be as high as 123, the maximum number for $N_{\text{token}}$ can be $123 \times 1.6 - (13 + 26) = 157.8$. Using speculative decoding (Leviathan et al., 2023) to speed up and increase the number of reasoning tokens for each chunk is also a valid option.

## F    UNSUCCESSFUL ATTEMPTS

We detail the unsuccessful attempts and negative findings observed in our experiments. Our aim is to guide future research by highlighting these pitfalls so that others can avoid them.

**LoRA Fine-tuning**    Initially, we tried to use LoRA (Hu et al., 2022) for fine-tuning. However, when fine-tuning TBS using LoRA, we found that the resulting model cannot generate useful reasoning. After fine-tuning, the performance on GSM8K is still around 35%, which is almost the same as the accuracy of GLM-4-Voice. By observing the outputs from the fine-tuned model, we observed that the model can generate outputs that satisfy the output format in the training data, i.e., reasoning and then interleaved text-speech tokens. However, the reasoning is seldom correct and can make very trivial mistakes. We suspect that this is because teaching GLM-4-Voice mathematical reasoning requires significant model parameter updates, making LoRA fine-tuning a bad choice.

**Fine-tuning only on Math Datasets**    We have tried to fine-tune only on the math datasets to teach the model reasoning. However, we found that the resulting model will treat every input query as a math question and list some equations to solve the question, even if the question simply asks "*Which city is located at the intersection of the Tigris and Euphrates rivers?*" This is possibly because the model overfits to the pattern in the training data and forgets how to solve other non-math questions. However, this can be easily solved by including diverse types of tasks in the training data, which is what we do in the end.

**Customized Attention Mask**    When training the STITCH-R model, we tried with a variant with customized attention masks. Precisely, we make the reasoning tokens unable to attend to the text tokens and speech tokens, while the text and speech tokens can attend to all types of tokens. The attention mask still remains causal. The rationale for this customized attention mask is to make the

**You will be given a user's question and two audio responses from voice assistants A and B. Your task is to assess which assistant is more responsive.**

The user asks a question, and then the voice assistant responds. Each audio clip from the voice assistants begins at the moment the user finishes speaking the question. Please play both clips in full and rate responsiveness on a 7-point scale. **Responsiveness** means how quickly and smoothly the assistant starts speaking after the user finishes (minimal silence or pauses). If a clip starts with silence, it may be because the assistant is "thinking"; please wait patiently.

- +3: A is more responsive than B
- +2: A is somewhat responsive than B
- +1: A is slightly responsive than B
- 0: A and B are equally responsive
- -1: B is slightly responsive than A
- -2: B is somewhat responsive than A
- -3: B is more responsive than A

⚠️ **Important: Audio May Start with Silence**

Some clips intentionally begin with silence while the assistant is "thinking." **Do not increase your volume during this silence.** When speech starts, a sudden loud burst could harm your hearing.

- **Start with a low volume.**
- **Set your volume using the calibration clip below** before you play the main experiments.
- **Use the following audio to calibrate your volume (this clip has no latency):**

▶ 0:00 / 0:18 ——————————— 🔊 ⋮

*Thank you for listening safely.*

**Please listen to the FULL audio before you make your judgement. If we found evidence that you did not listen to the full audio, we will reject your answer. We will reject the answers from those who are obviously spamming the task.**

**Question 1**

**A treasure hunter found a buried treasure chest filled with gems. There were 175 diamonds, 35 fewer rubies than diamonds, and twice the number of emeralds than the rubies. How many of the gems were there in the chest?**

**Voice Assistant A**

▶ 0:00 / 0:17 ——————————— 🔊 ⋮

**Voice Assistant B**

▶ 0:00 / 0:19 ——————————— 🔊 ⋮

**Which assistant is more responsive? (A or B)**
- ○ **+3: A is more responsive than B**
- ○ **+2: A is somewhat responsive than B**
- ○ **+1: A is slightly responsive than B**
- ○ **0: A and B are equally responsive**
- ○ **-1: B is slightly responsive than A**
- ○ **-2: B is somewhat responsive than A**
- ○ **-3: B is more responsive than A**

Figure 4: The interface of the human evaluation.

reasoning tokens feel like there is no *gap* between two chunks of reasoning tokens, since they cannot see the text and speech tokens that separate two chunks of reasoning tokens. We use the customized attention mask during training and inference. However, we found that the results on the math reasoning datasets are similar to the results of not using a customized attention mask. Considering that the customized attention mask requires some customization during training, complicates the inference procedure, and does not improve the performance, we do not recommend this configuration.

| Id | Config | Use Reasoning Train | Infer. | AddSub | MultiArith | SinglEq | SVAMP | GSM8K | Avg. |
|----|--------|-------|--------|--------|------------|---------|-------|-------|------|
| | | | | *Baselines* | | | | | |
| 1 | GLM-4-Voice | – | ✘ | 49.54 | 55.75 | 66.06 | 40.13 | 25.80 | 47.46 |
| 2 | No reasoning | ✘ | ✘ | 62.39 | 65.52 | 71.56 | 57.53 | 33.13 | 58.03 |
| 3 | Mix reasoning | ✔+✘ | ✘ | 62.39 | 77.59 | 85.32 | 60.87 | 36.49 | 64.53 |
| | | | | *Full Reasoning before Speaking* | | | | | |
| 4 | TBS | ✔ | ✔ | 70.64 | 79.31 | 84.40 | 71.57 | 60.69 | 73.32 |
| 5 | Mix reasoning | ✔+✘ | ✔ | 67.89 | 86.21 | 83.49 | 69.57 | 59.92 | 73.42 |
| | | | | *Proposed Methods: Simultaneous Thinking and Speaking* | | | | | |
| 6 | STITCH-R | ✔ | ✔ | 67.89 | 83.33 | 88.07 | 69.13 | 53.05 | 72.29 |
| 7 | STITCH-S | ✔ | ✔ | 73.39 | 84.48 | 81.65 | 66.78 | 52.18 | 71.70 |

Table 7: The performance on math reasoning datasets when evaluating the transcription of the speech output. We use Whisper-large-v3 Radford et al. (2023) to transcribe the audio. The *"use reasoning"* columns denote whether text reasoning is in the training data or is generated during inference.

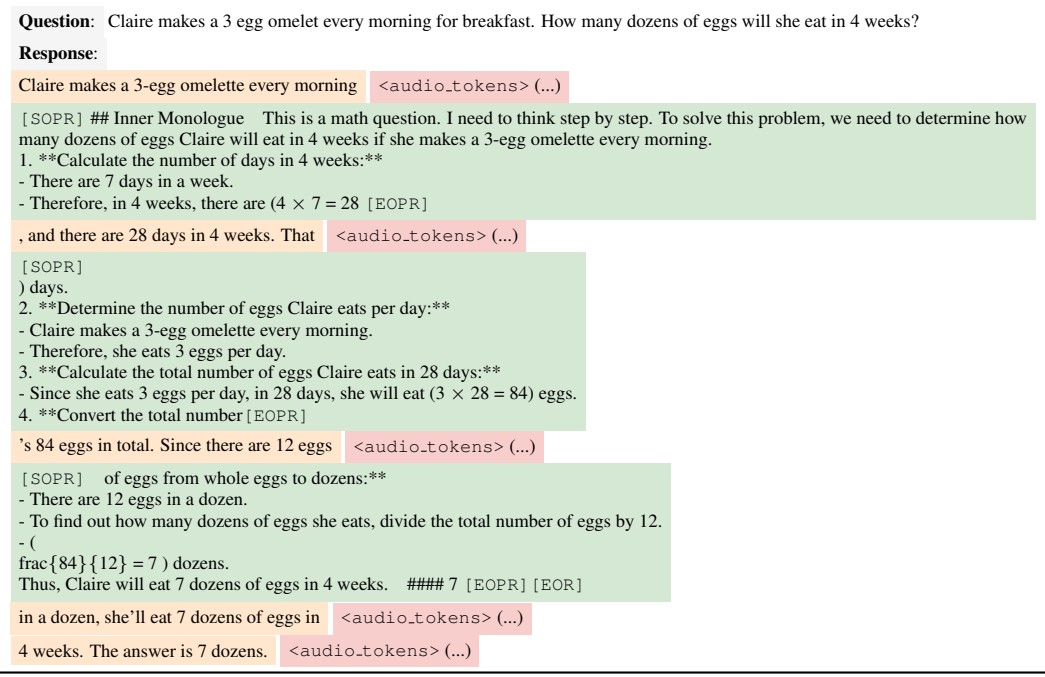

Table 8: An Example output from STITCH-S. The text reasoning tokens are highlighted in green; the text tokens are highlighted in orange; the audio tokens, which are omitted for simplicity, are highlighted in red. We remove and insert some newlines for formatting.

```
You are a helpful evaluation assistant.  You will be given a question, its ground
truth answer, and a prediction by an AI assistant.  Your job is to evaluate the
fluency of the answer into a single score.  The score you can choose from is 1, 2,
3, 4, 5.  Note that you **do not** need to judge the accuracy of the AI assistant's
prediction; you only need to consider how fluent the sentences are.

## Evaluation Criteria:
Fluency and Coherence (1{5):  This rating measures the quality of individual
sentences|are they well-written and grammatically correct?  It also covers the
overall organization and structure of the prediction.

## Output Format:
Please provide some reasoning about how you evaluate the prediction and explain what
factors affect your judgment.  Conclude your answer with 'Final verdict:  score',
where 'score' is an integer from [1,2,3,4,5].

## Question
{question}

## Ground Truth answer
{answer}

## Prediction by AI assistant
{prediction}

## Evaluation Task
How fluent and coherent is the prediction?  (On a scale of 1{5, with 1 being the
lowest)
```

Table 9: The prompt used to evaluate the fluency of the text tokens from the model with LLM-as-a-judge.

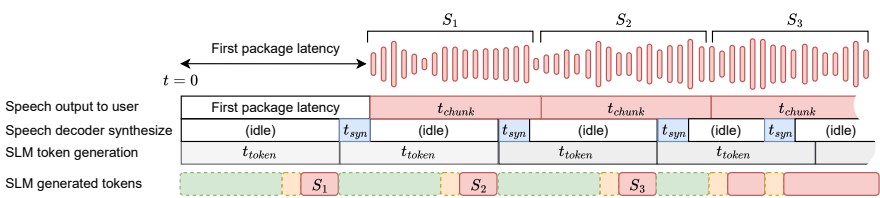

(a) An illustration of timing diagram with a smaller $t_{syn}$, the time for synthesizing $N_{\text{speech}}$ tokens in the the corresponding audio.

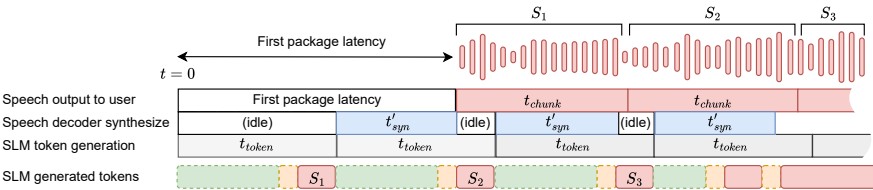

(b) An illustration of timing diagram with a larger $t'_{syn}$, the time for synthesizing $N_{\text{speech}}$ tokens in the the corresponding audio.

Figure 5: The above two figures show the same STITCH-R model with different $t_{\text{syn}}$. The upper one has a smaller $t_{\text{syn}}$, while the lower one has a larger $t_{\text{syn}}$. The figure is used to show that the speech decoder and the token generation of STITCH can be run in parallel. Consequently, as long as the $t_{\text{syn}} < t_{\text{chunk}}$, two audio chunks can be played seamlessly to the user. The difference in $t_{\text{syn}}$ only affects the first packet latency.

```
You will be given a math question and its inner reasoning about the math question
that is used to derive the final answer.  Your job is to summarize the reasoning
into a short and concise answer that is suitable to be spoken in dialogue.  You
should avoid using excessive equations and math notations but only briefly mention
how the result is calculated.  You don't need to restate each step and intermediate
results in the inner reasoning; just provide a brief overall summary.

## Example 1:
### Question ###
Natalia sold clips to 48 of her friends in April, and then she sold half as many
clips in May.  How many clips did Natalia sell altogether in April and May?
### Inner reasoning ###
I need to calculate the number of clips Natalia sold in April and May.  I am
given that she sold 48 in April, and I need to calculate how many she sold in May.
We know that she sold half as many in May, so 48 / 2 = 24.  She sold 24 in May.
Combining the answer, she sold 48 + 24 = 72 in April and May.
### Summarized spoken answer ###
In April she sold 48 clips and in May she sold half that amount, which is 24, so
altogether she sold 72 clips.

## Example 2:
### Question ###
Weng earns $12 an hour for babysitting.  Yesterday, she just did 50 minutes of
babysitting.  How much did she earn?
### Inner reasoning ###
We need to calculate how much she earned in 50 minutes.  We are given that she earns
12 per hour.  50 minutes corresponds to 50 / 60 = 5 / 6 hours.  As a result, she can
earn 12 * (5 / 6) = 10.  The answer is 10.
### Summarized spoken answer ###
She worked 50 minutes, which is five-sixths of an hour at $12 per hour, so she
earned $10.

## Example 3:
### Question ###
Betty is saving money for a new wallet which costs $100.  Betty has only half of
the money she needs.  Her parents decided to give her $15 for that purpose, and her
grandparents twice as much as her parents.  How much more money does Betty need to
buy the wallet?
### Inner reasoning ###
We need to calculate how much more money Betty needs.  She has saved $50, her
parents gave $15, and grandparents gave $30, so she has $95 in total and needs $5
more.  The answer is 5.
### Summarized spoken answer ###
She has $95 so far from her own savings and gifts, and since the wallet costs $100,
she still needs $5.

# Now it is your turn
### Question ###
{question}
### Inner reasoning ###
{reasoning}
### Summarized spoken answer ###
```

Table 10: The prompts used to rewrite the answers in the dataset into the spoken answer for the math reasoning dataset in the training data. The {question} is the placeholder of the question; the {reasoning} will be replaced by the reasoning, which is the original answer in the training data.

```
You will be given a factual question and its answer.  It is possible that there are
multiple answers to the question, and we are only providing one for simplicity.
Your job is to generate some inner thinking to derive the answer.  Imagine that
you are a human and you heard somebody ask you that question, and you should think
before responding to the question in speech.  Your inner thinking can include
rephrasing the question into a better way for understanding, and clarifying any
possible ambiguity, retrieving the correct answer and some relevant information.
Your inner reasoning should include the short answer.  You should not say "I know
the answer" without explicitly saying the answer.  You should not say "I need more
information" and end your reasoning since this is not helpful at all.

## Example 1
## Question:
What color was John Wilkes Booth's hair?
## Answer
jet-black
## Inner thinking
I think the user is asking about John Wilkes Booth who assassinated United States
President Lincoln on April 14, 1865.  His hair color is jet-black.  Got the answer
and ready to provide it to the user.

## Example 2
## Question:
Can you make and receive calls in airplane mode?
## Answer
No
## Inner thinking
Airplane mode is a setting on smartphones and other portable devices that suspends
all radio-frequency transmissions.  So you can't make or receive calls while it's on.
The answer is no.

## Example 3
## Question:
Who played "Will Munson" on "As the World Turns"?
## Answer
Jesse Lee Soffer
## Inner thinking
The user likely means the CBS soap "As the World Turns." The character Will Munson
was most recently portrayed by Jesse Lee Soffer.  Ready to speak the answer.

# Now it is your turn
## Question:
{question}
## Answer
{answer}
## Inner thinking
```

Table 11: Prompts used to generate reasoning for NQ and TriviaQA in the training data.

```
You will be given a factual question and its short answer.  We will also provide you
some inner reasoning that leads to the answer.
It is possible that there are multiple answers to the question, and we are only
providing one for simplicity.
Your job is to rewrite the short answer such that it is appropriate for spoken form.
When asked a factual question, humans will think about the question in their mind
and come up with an answer that summarizes their inner reasoning in a concise and
clear response that answers that question and keeps only the relevant information.
Your job is to form this spoken answer based on the question, the inner reasoning,
and the short answer.

## Example 1
## Question:
What color was John Wiles Booth's hair?
## Inner thinking
I think the user is asking about John Wilkes Booth who assassinated United States
president Lincoln on April 14, 1865.  His hair color is jet-black.  Got the answer
and ready to provide it to the user.
## Short answer
jet-black
## Spoken answer
You mean the one who killed Lincoln?  His hair color was jet-black.

## Example 2
## Question:
Can you make and receive calls in airplane mode?
## Inner thinking
Airplane mode is a setting available on smartphones and other portable devices.
When activated, it suspends all radio-frequency transmissions, disabling voice and
data services.  So you can't make or receive calls while it's on.
## Short answer
no
## Spoken answer
Since airplane mode disables all signal transmissions, you cannot make or receive
calls in airplane mode.

## Example 3
## Question:
Who played "Will Munson" on "As the World Turns"?
## Inner thinking
The user likely means the CBS soap opera "As the World Turns." The character Will
Munson was most recently portrayed by Jesse Lee Soffer.
## Short answer
Jesse Lee Soffer
## Spoken answer
It's Jesse Lee Soffer|he played Will Munson on that soap opera.

# Now it is your turn
## Question:
{question}
## Inner thinking
{reasoning}
## Short answer
{answer}
## Spoken answer
```

Table 12: Prompts used to rewrite the original answer into spoken forms for NQ and TriviaQA.

```
# Task Overview:  Inner Monologue Creation
A voice assistant receives a user query or instruction and produces an answer.  We
now want to come up with a brief \inner monologue" that the assistant might have
before speaking.
User Question/Instruction:  {question}
Assistant Answer:  {answer}

# Task Introduction
Generate a concise inner thinking that the assistant could have had before speaking.
Your inner monologue should include understanding of the user's question or
instruction and then a draft answer.
Make sure the inner monologue will lead to the assistant's provided answer.

# Examples

## Example 1
User Question/Instruction:  How can I stop being afraid of the dark?
Assistant Answer:  It's completely normal to feel afraid of the dark, and there are
steps you can take to feel more comfortable.  Start by gradually exposing yourself
to the dark in small, manageable steps.  You can use a nightlight as a transitional
tool.  Establish a bedtime routine that makes you feel relaxed, like reading a book
or listening to calming music.  Breathing exercises or meditation can help reduce
anxiety.  Rearrange your bedroom to make it feel safer and more inviting.  Remind
yourself that the fear is often about not knowing what's there, rather than actual
dangers.  Talking to someone about your fears can also be really helpful.
Inner Monologue:  The user wants to stop being afraid of the dark.  I will first
acknowledge that it's normal to feel fear, then suggest gradual exposure, a
comforting bedtime routine, anxiety-reducing exercises, and talking to someone.

## Example 2
User Question/Instruction:  Can you count down from 100 by subtracting 7 each time?
Assistant Answer:  Yes, I can.  Help you with counting down from 100 and minus 7
each time.  Here we go:  100, 93, 86, 79, 72, 65, 58.  Do you need more?
Inner Monologue:  The user wants a countdown by sevens from 100.  I'll compute
100-7=93, then subtract 7 each step until I have a sequence.  I'm ready to present
the numbers.

## Example 3
User Question/Instruction:  What are some interesting things I can mix into my
scrambled eggs?
Assistant Answer:  Scrambled eggs can be a delightful canvas for a variety of
ingredients!  You can try adding cheese like cheddar, feta, or Swiss for extra
creaminess.  Fresh herbs such as chives, parsley, or dill can bring a burst of
flavor.  Diced vegetables like bell peppers, tomatoes, spinach, or mushrooms can
add texture and nutrition.  For protein, consider mixing in smoked salmon, ham,
or crumbled bacon.  If you enjoy a bit of heat, a dash of hot sauce or chopped
jalapeños can be great.  Even a bit of sour cream or cream cheese can make the eggs
extra creamy.  Enjoy experimenting to find your perfect blend!
Inner Monologue:  The user is asking for mix-ins for scrambled eggs.  I'll recall
common add-ins like cheeses, herbs, vegetables, proteins, and creams, then frame
them as suggestions.

# Your Task
Generate the inner monologue for this instruction/answer pair.
User Question/Instruction:  {question}
Assistant Answer:  {answer}
Inner Monologue:  (provide the concise inner monologue only)
```

Table 13: The prompt used to generate the reasoning for VoiceAssistant400K.

