# OpenReview forum: "STITCH: Simultaneous Thinking and Talking with Chunked Reasoning for Spoken Language Models"
_ICLR.cc/2026/Conference — ICLR 2026 Poster_

### Official Review · Reviewer_TVGk · 2025-10-25

**Soundness:** 3
**Presentation:** 3
**Contribution:** 3
**Rating:** 6
**Confidence:** 5

**Summary:**

This paper introduces STITCH, a Spoken Language Model (SLM) that performs internal reasoning alongside speech generation, theoretically without increasing interaction latency. STITCH generates speech chunk-by-chunk, with each chunk containing reasoning tokens, text tokens, and speech tokens. Experimental results demonstrate that this method improves reasoning performance compared to baseline approaches.

**Strengths:**

- The idea of chunk-wise reasoning alongside speech generation is novel.
- The investigation into optimal reasoning chunk length and the use of reasoning from other models adds further novelties.
- Experimental results clearly show the effectiveness of the proposed method.

**Weaknesses:**

- The paper does not compare STITCH against thinker-talker models. The substantial drop in reasoning performance observed in interleaved models (relative to text-only LLMs) often stems from the interleaved LLM's need to generate both text and speech tokens. In contrast, the thinker component in thinker-talker models generates only text, addressing this limitation. Although the authors mention that thinker-talker models are harder to fine-tune—which may imply challenges in adapting STITCH to such architectures—it would be fairer to include thinker-talker models in the comparisons.
- Evaluation is conducted solely on text outputs. While this is somewhat acceptable, it would be more robust to also evaluate directly on speech outputs, since the performance gap between text and speech evaluations can depend on the backbone LLM’s text-speech alignment capabilities.
- There is an inconsistency in the introduction section. Lines 084-087 state that STITCH-S cannot generate reasoning by design, but it actually generates text reasoning—it simply does not generate text reasoning in the first chunk. Am I missing something here?
- Statistical analysis should be performed to assess whether there are significant differences in responses among TBS, STITCH-R, STITCH-S, etc.

**Questions:**

N/A.

---

> ### Author Response · Authors · 2025-11-20
> **Response to Reviewer TVGk**
>
> Thank you for your valuable reviews. We respond to the weaknesses as follows.
>
> > The paper does not compare STITCH against thinker-talker models. The substantial drop in reasoning performance observed in interleaved models (relative to text-only LLMs) often stems from the interleaved LLM's need to generate both text and speech tokens. In contrast, the thinker component in thinker-talker models generates only text, addressing this limitation. Although the authors mention that thinker-talker models are harder to fine-tune—which may imply challenges in adapting STITCH to such architectures—it would be fairer to include thinker-talker models in the comparisons.
>
> We kindly ask the reviewer to refer to the general response about the discussions on different SLMs.
>
> > Evaluation is conducted solely on text outputs. While this is somewhat acceptable, it would be more robust to also evaluate directly on speech outputs, since the performance gap between text and speech evaluations can depend on the backbone LLM’s text-speech alignment capabilities.
>
> **In the last paragraph in Section 4.2, we discuss how we evaluate the speech output**. Precisely, we run ASR to get the transcription of the speech output and then evaluate the text transcription. The results, as shown in Table 6 in the Appendix, reveal that **all observations we found in the text outputs are consistent with the results on the speech output**. However, the speech output exhibits a slight performance drop of around 3% to 6%, which may be attributed to imperfect ASR transcription.
>
> > There is an inconsistency in the introduction section. Lines 084-087 state that STITCH-S cannot generate reasoning by design, but it actually generates text reasoning—it simply does not generate text reasoning in the first chunk. Am I missing something here?
>
> Our original sentence is, ”*STITCH-S, a variant of STITCH, has the same latency as the baseline models that cannot generate unspoken reasoning by design*“. Here, the one that cannot generate reasoning is the **baseline model,** i.e., the “no reasoning” baseline in Section 4.3. We will tweak the wording here to avoid possible confusion. We thank the reviewer for raising this point.
>
> > Statistical analysis should be performed to assess whether there are significant differences in responses among TBS, STITCH-R, STITCH-S, etc.
>
> We use McNemar’s test(two-sided, $\\alpha \= 0.05$), which is to justify our claims in Section 5 to say one model is significantly better than another. We have the following observation:
>
> 1) TBS, STITCH-R, and STITCH-S are significantly better than the no-reasoning baseline, supporting our claims in Section 5\.
> 2) Additionally, we check if STITCH-R is significantly better than STITCH-R. We found that on all math datasets except AddSub, the advantage of STITCH-R over STITCH-S is statistically significant. Although the accuracy difference between the two models is small, it is statistically significant.
>
> We have incorporated the clarifications and revisions above, and we believe they address the concerns raised. If any questions remain, please let us know, and we are happy to provide further details. We are looking forward to learn your feedback on our responses.

---

> > ### Comment · Reviewer_TVGk · 2025-11-23
> >
> > I appreciate the authors' efforts in the rebuttal. I would like to clarify one point: when I said "The paper does not compare STITCH against thinker-talker models," I did not mean that the authors should also fine-tune the thinker-talker models. My actual intention is that models like Qwen2.5-Omni should serve as the no-reasoning baseline in the benchmark comparison. Since thinker-talker models address the limitations of the interleaved models, they should at least serve as no-reasoning baselines to ensure fair comparisons.

---

> ### Author Response · Authors · 2025-11-23
> **Re: Official Comment by Reviewer TVGk**
>
> Thank you for your response. We apologize for the misunderstanding regarding your original review.
>
> To respond to your original meaning, we think that **using Qwen-2.5-Omni may not be a reasonable baseline because the Qwen-2.5-Omni has undergone a supervised fine-tuning (SFT) stage that might deliberately include a math reasoning dataset.** (The paper of Qwen-2.5-Omni has not disclosed any details related to SFT, so we are also not sure what exact data they really used, and we are not sure if the scale matches the data scale we use. We speculate they include math data based on the final performance of Qwen-2.5-Omni.) Using Qwen-2.5-Omni as the baseline to compare with our models fine-tuned using GLM-4-Voice will make the comparison less meaningful **since this includes another confounder: the SFT data used.** GLM-4-Voice explicitly says that during the SFT stage, "*code and math-related content are excluded to focus on conversational material suitable for spoken interactions*. **Therefore, it is unclear whether the thinker-talker models, as a modeling architecture, truly address the limitation of the interleaved models, or whether they are simply trained on different SFT datasets.**
>
> For the sake of completeness, we still report the performance of Qwen-2.5-Omni 7B on the math reasoning datasets we use. The results here are evaluated from the thinker output of Qwen-2.5-Omni, which matches the evaluation setting in Table 1(a) in the paper. However, we must emphasize that **the following results should not be compared with the results in our paper since the models are possibly trained on completely different SFT datasets.**
>
>
> |  | AddSub | SinglEq | SVAMP | MultiArith | GSM8K |
> | :---: | :---: | :---: | :---: | :---: | :---: |
> | Qwen-2.5-Omni  | 92.8 | 97.4 | 90.3 | 98.0 | 85.4 |
>
> ---
>
> As a personal note not directly related to this paper, I would greatly appreciate it if community members could conduct a truly fair comparison between the interleaved SLMs and the thinker-talker SLMs, as this can help the practitioners to understand if there is really a superiority between the two architectures.
>
>
> ----
> ----
> ----
>
> ### Again, thank you for your engagement in the author-reviewer discussion period. Do these and previous responses resolve your concerns on these points (including the four weaknesses in your initial review)? We’re happy to add any further clarification you recommend.

---

### Official Review · Reviewer_Yahr · 2025-10-29

**Soundness:** 3
**Presentation:** 3
**Contribution:** 2
**Rating:** 4
**Confidence:** 3

**Summary:**

This study introduces STITCH, a method designed to incorporate reasoning into speech by interleaving it within the existing Speech-Language Model (SLM) framework—specifically building upon GLM-4-Voice, one of the text–speech interleaving architectures (e.g., GLM-4-Voice, VITA-Audio). The authors propose two variants, -R and -S, depending on whether reasoning appears at the very beginning or after the first text–speech chunk.
The proposed approach is optimized based on the A100 + vLLM configuration. Compared to the baseline method, in which GLM-4-Voice performs “thinking” before speaking, STITCH demonstrates comparable performance on mathematical reasoning tasks and achieves performance on general conversational tasks that is similar to the original GLM-4-Voice.

**Strengths:**

**[S1]** The paper is written clearly and is easy to follow. The operational mechanism of the proposed method is straightforward to understand.

**[S2]** Notably, the model exhibits comparable results on mathematical tasks compared to approaches that explicitly perform reasoning with reduced latency.

**Weaknesses:**

While the study presents an interesting direction, the scope of its **novelty and generalizability appears somewhat limited**. The following points are offered as considerations rather than criticisms:

**[W1]** The reported optimization is based on the **A100 + vLLM** setting, which may limit the applicability of the results. It remains uncertain whether the proposed approach would generalize well to limited hardwares, larger models, or alternative architectures, such as those that jointly optimize a separate decoder (or talker) during inference rather than interleaving, or models that predict codecs (e.g., *moshi*). In addition, the interleaving method applied in this study appears to extend the existing text–speech interleaving approach, already performed in models such as GLM-4-Voice and VITA-Audio, to the reasoning component, without introducing additional architectural or methodological considerations specific to reasoning itself.

- **[Q1]** To what extent do the reported gains persist across different hardware budgets, model scales, and architectural variants (e.g., non-interleaving pipelines with a joint talker/decoder, or codec-predictive systems such as moshi)?

**[W2]** The dataset used comprises general dialogue, mathematical, and knowledge-intensive questions generated through **GPT-4o**. It is possible that employing reasoning-specialized models, such as **Qwen3-235B** or **DeepSeek**, for path construction might have yielded broader improvements, particularly in the non-reasoning tasks presented in Table 1. The comparable performance to the baseline GLM-4-Voice on general tasks therefore leaves some ambiguity regarding how the reasoning path contributes in those contexts.

- **[Q2]** Does the choice of path-generating model (e.g., GPT-4o vs. reasoning-oriented models) materially affect downstream performance, especially on non-reasoning tasks (Table 1)?

**[W3]** The paper does not provide an analysis or guarantee as to whether the reasoning path consistently concludes earlier in the 100 / 13 / 26 configuration, nor whether reasoning reliably precedes the corresponding text segments. Given that human reasoning typically precedes linguistic expression, a more explicit discussion or examination of this alignment could further strengthen the study.

- **[Q3-1]** Does reasoning precede and complete before corresponding text/speech segments, and does early termination of reasoning path occur in the 100 / 13 / 26 configuration?

- **[Q3-2]** If the reasoning segment tends to finish much earlier than the corresponding text, could the authors examine (1) whether adjusting the reasoning chunk length during training so that it depletes at a pace similar to the text segment, and (2) independently, enforcing a structure where reasoning always precedes the textual response (to prevent the model from answering first and reasoning afterward), would lead to different experimental results?

**Questions:**

See weaknesses

---

> ### Author Response · Authors · 2025-11-20
> **Response to Reviewer Yahr (1/3)**
>
> First, we thank the reviewer for noting that the comments are more of a consideration rather than a criticism. We address the considerations as follows.
>
> > \[W1\] The reported optimization is based on the A100 \+ vLLM setting, which may limit the applicability of the results. It remains uncertain whether the proposed approach would generalize well to limited hardwares, larger models, or alternative architectures, such as those that jointly optimize a separate decoder (or talker) during inference rather than interleaving, or models that predict codecs (e.g., moshi). In addition, the interleaving method applied in this study appears to extend the existing text–speech interleaving approach, already performed in models such as GLM-4-Voice and VITA-Audio, to the reasoning component, without introducing additional architectural or methodological considerations specific to reasoning itself.
>
>
>
> We would like to highlight that introducing a reasoning stream into SLMs is non-trivial and, to our knowledge, has not been explored in prior works. In previous interleaved SLMs, including GLM-4-Voice and VITA-Audio, all text tokens are ultimately meant to be spoken. In contrast, STITCH explicitly separates reasoning tokens, which are never spoken, from spoken text tokens, and designs the interleaving schedule so that the model can perform internal reasoning without significantly increasing end-to-end latency. This separation between **“silent”** reasoning and **spoken** content is a core conceptual difference from prior interleaved SLMs, and we believe it is a main contribution of our paper. To the best of our knowledge, **STITCH is the first demonstration that reasoning can be integrated into SLMs while preserving streaming latency**. We hope the reviewer can consider the contribution to introducing reasoning into SLM as a strength of our paper.
>
> > \[Q1\] To what extent do the reported gains persist across different hardware budgets, model scales, and architectural variants (e.g., non-interleaving pipelines with a joint talker/decoder, or codec-predictive systems such as moshi)?
>
> We first discuss different hardware budgets. While all our experiments are conducted on A100 using vLLM, **our analysis in Section 7 considers the case when the hardware has a smaller token throughput**. Precisely, we decrease $N\_{reason}$ during inference time to simulate the case when we have slower hardware, where we cannot generate $N\_{reason}=100$. The results in Section 7 (and Figure 3(a) in the appendix) show that with a lower $N\_{reason}$, STITCH can still perform reasonably well most of the time. Precisely, **when we can set $N\_{reason}=80$, which is only 80% of the token throughput on A100, the performance of STITCH only drops by about 10%, while still significantly outperforming the no-reasoning baseline**.
>
> Accelerating the inference speed on diverse hardware is more of an engineering problem and is not the main focus of this paper, which mainly focuses on proposing a new inference method. However, in Line 1051 of Appendix E, we mention that more advanced speed-up methods, such as speculative decoding, can be used to increase the reasoning token generation speed. Additionally, more extreme quantization can also reduce the inference latency. We leave optimizing the inference speed to practitioners while pointing out the above possible directions. We will add the above discussion to the paper.
>
> We kindly ask the reviewer to refer to the general response about the discussions on different SLMs.

---

> > ### Author Response · Authors · 2025-11-20
> > **Response to Reviewer Yahr (2/3)**
> >
> > > \[W2\] The dataset used comprises general dialogue, mathematical, and knowledge-intensive questions generated through GPT-4o. It is possible that employing reasoning-specialized models, such as Qwen3-235B or DeepSeek, for path construction might have yielded broader improvements, particularly in the non-reasoning tasks presented in Table 1\. The comparable performance to the baseline GLM-4-Voice on general tasks therefore leaves some ambiguity regarding how the reasoning path contributes in those contexts.
> >
> > > \[Q2\] Does the choice of path-generating model (e.g., GPT-4o vs. reasoning-oriented models) materially affect downstream performance, especially on non-reasoning tasks (Table 1)?
> >
> > ### Short Answer
> >
> > In short, we have tried to use reasoning-oriented models (Deepseek-r1) to generate the reasoning for non-reasoning tasks, but we do not find the reasoning paths to be significantly different from those generated by GPT-4o. We eventually chose GPT-4o for its better performance on non-reasoning tasks and some practical considerations.
> >
> > ### Long Answer
> >
> > In the initial stage of this project, we also consider other alternatives to generate the reasoning, including reasoning models like Deepseek-r1. To use Deepseek-r1 to generate the reasoning for the non-reasoning datasets like factual QA (e.g., NaturalQuestion), a straightforward way to provide Deepseek-r1 with the question, and use the internal thinking of Deepseek-r1 as the reasoning path for STITCH. In our preliminary experiments, we found that the reasoning from Deepseek-r1 is quite similar to the reasoning generated by GPT-4o when prompted with the instruction in Table 10\. For example, the following is the deep thinking from Deepseek-r1 when asked the question “*when did richmond last play in a preliminary final?*” from NaturalQuestion:
> >
> > \`Hmm, the user is asking about Richmond's most recent preliminary final appearance. This is a straightforward factual question about AFL history.
> > I recall Richmond had a strong period recently but have declined since 2022\. The key is to pinpoint their last prelim final year first \- that would be 2020 during their three-premiership era.
> > Should provide the exact date and opponent for that match, then briefly contextualize it within their recent timeline. The response should be concise but include the essential details: year, date, opponent, result, and what came after.
> > Better verify the 2020 preliminary final details quickly \- yes, it was against Port Adelaide on October 16th during the COVID-affected season. The user might also appreciate knowing this was part of their back-to-back premierships.\`
> >
> > The above reasoning chain from Deepseek-r1 closely mirrors what we prompt GPT-4o to do: rephrasing the question into a better way for understanding, clarifying any possible ambiguity, and retrieving the correct answer and some relevant information. In short, **prompting GPT-4o gives similar results to directly asking Deepseek-r1 on non-reasoning tasks,**  so using GPT-4o and Deepseek-r1 seems equally good.
> >
> > Eventually, we select GPT-4o over Deepseek-r1 to generate the reasoning for the non-reasoning datasets for the following three reasons:
> >
> > 1. **GPT-4o has a better performance on factual knowledge compared to Deepseek-r1**. Precisely, GPT-4o’s accuracy on SimpleQA is 8.2% higher than Deepseek-r1, according to Deepseek-r1’s technical report.
> > 2. **Prompting GPT-4o to generate the reasoning enables more control over the reasoning path**, as the output from GPT-4o is specifically aligned to follow the user instruction. In contrast, we cannot really control what Deepseek-r1 generates in its reasoning (those enclosed in the \<think\> and \</think\> tokens). This is mainly because, for reasoning models, the thinking process is not specifically monitored.
> > 3. GPT-4o is easier and faster to use for us

---

> ### Author Response · Authors · 2025-11-20
> **Response to Reviewer Yahr (3/3)**
>
> > \[W3\] The paper does not provide an analysis or guarantee as to whether the reasoning path consistently concludes earlier in the 100 / 13 / 26 configuration, nor whether reasoning reliably precedes the corresponding text segments. Given that human reasoning typically precedes linguistic expression, a more explicit discussion or examination of this alignment could further strengthen the study.
>
> > \[Q3-1\] Does reasoning precede and complete before corresponding text/speech segments, and does early termination of reasoning path occur in the 100 / 13 / 26 configuration?
>
> > \[Q3-2\] If the reasoning segment tends to finish much earlier than the corresponding text, could the authors examine (1) whether adjusting the reasoning chunk length during training so that it depletes at a pace similar to the text segment, and (2) independently, enforcing a structure where reasoning always precedes the textual response (to prevent the model from answering first and reasoning afterward), would lead to different experimental results?
>
> Thank you for your question. We will include the following discussion in Section 5 in the revision.
>
> ### \[A3-1\]
> In short, our training ensures that the reasoning path always concludes before the text/speech segment. This is because in the training data, we always ensure that the reasoning concludes earlier, as specified in the last paragraphs in Section 3.2 and 3.3. As a result, the model will learn that the reasoning must finish before the text/speech segments. During inference, the model always completes the reasoning before processing the text/speech segments.
>
> ### \[A3-2\]
>
> \- **The reasoning segments indeed tend to finish much earlier than the text**. This can be seen from the statistics in Table 2, left. For example, on GSM8K, STITCH-R has on average $322.40/100 \=3.22$ reasoning chunks, while the average number of text chunks is $74.36/13=5.72$.
>
> \- We do not specifically design the reasoning chunk length during training to ensure that the reasoning depletes at a pace similar to the text segment. If we want to do so, the tokens in a single reasoning chunk need to be increased beyond 100, the default $N\_{reason}$, and the duration of a single audio chunk will not be sufficient to cover the time to generate a chunk of reasoning/text/speech tokens with $N\_{reason} \> 100$.
>
> \- **STITCH-R/S has reasoning preceding the textual response in about 95% of the cases at inference time, i.e., the model reason first and then answer later**. While in the training data, we do not strictly enforce the reasoning to always precede the text to prevent the model from speaking faster than it thinks, the choice of $N\_{reason}=100$ is typically sufficient to make the reasoning precede the textual response. In Appendix B, we investigate whether reason chunks anticipate future responses or are merely post-hoc justifications of previous textual responses.  We use GPT-4o as a judge, give 4o the full interleaved reasoning-text chunks generated by STITCH-R/S, and ask GPT-4o to determine whether the thinking is only a post-hoc justification of the previous text chunks. We find that in 95% of cases, the reasoning chunk is not the post-hoc explanations of prior response but rather the anticipation of future text chunks, meaning that the reasoning precedes the textual response most of the time.
>
> We have incorporated the clarifications and revisions above, and we believe they address the concerns raised. If any questions remain, please let us know, and we are happy to provide further details. We are looking forward to learn your feedback on our responses.

---

> > ### Comment · Reviewer_Yahr · 2025-11-20
> > **Additional question**
> >
> > Thank you for your kind answer.
> >
> > Is there any performance difference between the 95% of cases where the reasoning path appears first and the remaining 5% where the order is reversed?
> >
> > For example, does performance improve in both cases compared to when there is no reasoning path at all, but the improvement is larger in the 95% of cases where the reasoning path comes first?
> > Or does performance degrade in the cases where the reasoning path appears after the text response? I’m curious about the overall pattern of these results.

---

> > > ### Author Response · Authors · 2025-11-20
> > > **Re: Additional question**
> > >
> > > Thank you for your very quick follow-up question. This is really an interesting question. We analyze our experiment results, and **we do not find a significant difference in the accuracy of the two groups**. Precisely, for STITCH-R on GSM8K, the accuracy on the samples where the reasoning is always faster than the speech is 58.82, while the samples whose textual response is faster than the reasoning have an average accuracy of 58.69.
> > >
> > > We think this is because, although sometimes the textual response is faster than the reasoning, as shown in the second sample in Table 5 (discussed in Section 6), the reasoning already provides sufficient information and intermediate calculation for the textual response to come up with a result that has not been mentioned in prior reasoning. Consequently, the textual response can get faster than the reasoning without a performance drop.
> > >
> > > If you have any questions, please let us know, and we will be happy to provide further details. We look forward to learning your feedback on this and previous responses.

---

> > > > ### Comment · Reviewer_Yahr · 2025-11-21
> > > >
> > > > I'm sorry for the late reply; it was nighttime in my time zone, so I couldn’t check your message earlier.
> > > >
> > > > I still have some concerns, since the methodology is, in a sense, an extension of the existing text-speech interleaving approach. However, as you mentioned, the introduction and consideration of “thinking” within SLMs is a distinguishing factor, and I was also able to review the results of your additional experiments.
> > > >
> > > > Therefore, based on this, I will adjust my score accordingly. Again, I apologize for the delayed response.

---

> > > > > ### Author Response · Authors · 2025-11-21
> > > > > **Re: Official Comment by Reviewer Yahr**
> > > > >
> > > > > Thank you for your response and reconsideration of your original evaluation in light of our feedback.
> > > > > And your response is not late at all! It's much faster than we expected. Thank you for such a fast response. We are really happy about the multiple-round interaction and the interesting questions you raised.

---

### Official Review · Reviewer_G3Jr · 2025-11-01

**Soundness:** 3
**Presentation:** 3
**Contribution:** 3
**Rating:** 6
**Confidence:** 3

**Summary:**

The paper proposes STITCH, a generation method that alternates between unspoken reasoning chunks and spoken response chunks. It aims to significantly reduce the latency between reasoning generation and spoken response. Experimental results on both reasoning (e.g., math) and non-reasoning datasets, such as TriviaQA, show that STITCH performs on par with or better than baselines that either reason before speaking or do not perform reasoning at all.

**Strengths:**

- The authors introduce three ways of integrating reasoning into spoken language models: Thinking Before Speech (TBS), Simultaneous Thinking and Talking with Reasoning First (STITCH-R), and Simultaneous Thinking and Talking with Speaking First (STITCH-S). The methodology is clearly described, and Figure 2 effectively visualizes the differences between these methods.
- The analysis is good. The authors report performance while varying the length of reasoning chunks during inference and analyze the number of tokens used in the reasoning process. They also conduct experiments using different reasoning models to study how reasoning quality affects spoken responses.
- Experiments are conducted on both reasoning-oriented and non-reasoning datasets, making the experiments comprehensive.

**Weaknesses:**

- Based on the performance tables (Table 1-(a) and 1-(b)), there is no clear winner that consistently outperforms all other baselines. On math datasets, STITCH-R and STITCH-S show mixed performance across models, sometimes performing significantly worse than TBS (e.g., TBS 64.94, STITCH-R 58.70, STITCH-S 56.72). Similarly, on non-reasoning datasets, STITCH-R and STITCH-S perform inconsistently relative to other baselines. The paper does not clearly explain the reasons behind these trends.
- Following the previous point, there are no clear guidelines on when to use STITCH-R versus STITCH-S.
- The main motivation for reducing interaction latency is to improve user experience while maintaining response quality. However, the paper does not include any user-centric evaluation (e.g., human preference, perceived responsiveness, or satisfaction).
- It is unclear how STITCH handles cases where reasoning takes longer than speech generation. Would the model introduce pauses until reasoning is completed? Such pauses might make the users feel more inconvenient to use it.

**Questions:**

In STITCH-S, reasoning occurs after the response. In this case, the reasoning could be merely post-hoc justification. Why does this variant outperform baselines that reason before speaking or models without reasoning?

---

> ### Author Response · Authors · 2025-11-20
> **Response to Reviewer G3Jr (1/2)**
>
> We thank the reviewers for the detailed and positive review. We respond to your reviews as follows.
>
> > Based on the performance tables (Table 1-(a) and 1-(b)), there is no clear winner that consistently outperforms all other baselines. On math datasets, STITCH-R and STITCH-S show mixed performance across models, sometimes performing significantly worse than TBS (e.g., TBS 64.94, STITCH-R 58.70, STITCH-S 56.72). Similarly, on non-reasoning datasets, STITCH-R and STITCH-S perform inconsistently relative to other baselines. The paper does not clearly explain the reasons behind these trends.
>
> The performance in Table 1(a) is not used for showing which model is the best, but rather to show how different models trade off between performance and latency. **Relevant discussions are presented in Section 5\.**
>
> For the reasoning tasks, the main takeaway is that:
>
> 1. TBS is the best, but with the highest latency (First paragraph)
> 2. STITCH-R and STITCH-S exhibit performance slightly inferior to TBS on average; however, STITCH has a significantly lower latency. On a more challenging dataset like GSM8K, the accuracy drop becomes more observable. (Second and third paragraph)
>
> On non-reasoning datasets, all models, including reasoning and non-reasoning models, perform equally well, showing that reasoning does not harm the performance on tasks that do not need reasoning (Fourth paragraph)
>
> > Following the previous point, there are no clear guidelines on when to use STITCH-R versus STITCH-S.
>
> We thank the reviewer for bringing this to our attention, and we added relevant discussions in Section 5 in the revision. While STITCH-R and STITCH-S achieve similar performance on reasoning and non-reasoning datasets on average, STITCH-R still performs slightly better than STITCH-S on reasoning tasks, especially on more challenging datasets like GSM8K. However, STITCH-R generates an additional reasoning chunk before generating the speech response, resulting in STITCH-R having a higher latency (around 1.25 seconds more on A100). As a result, when such a latency is unacceptable, use STITCH-S instead of STITCH-R. However, if the task performance is more important while the 1.25s latency is acceptable, choose STITCH-R.
>
> > The main motivation for reducing interaction latency is to improve user experience while maintaining response quality. However, the paper does not include any user-centric evaluation (e.g., human preference, perceived responsiveness, or satisfaction).
>
> We are grateful to the reviewer for this feedback. Following the reviewer’s suggestion, we conduct a human evaluation on the responsiveness of different models and add it to Section 5\. We present the human evaluator with a question from the user and two audio responses from different models, asking them to compare the responsiveness of the two models on a seven-point scale, where \+3 indicates that model A is more responsive than model B and \-3 indicates that model B is more responsive than model A. To simulate the first package latency that the user may experience, each audio includes a silence in the beginning, accounting for the prefilling and decoding time of the SLM and the speech synthesis latency. We compare the following pairs: TBS / STITCH-R, TBS / STITCH-S, STITCH-R / STITCH-S, and STITCH / no-reasoning. For each pair, we use the same 100 questions from GSM8K, and we recruit 3 human evaluators to judge the responsiveness of a pair of models. More details on human evaluation are included in Appendix D in our revision.
>
> The results are presented in the table below, and we summarize the results as follows:
> 1\. STITCH-R and STITCH-S are both more responsive than TBS.
> 2\. STITCH-S is more responsive than STITCH-R.
> 3\. STITCH-S and no-reasoning are equally responsive.
>
> | Model A vs Model B | STITCH-R vs TBS | STITCH-S vs TBS | STITCH-R vs STITCH-S | STITCH-S vs no-reasoning |
> | :---: | :---: | :---: | :---: | :---: |
> | Mean score | 1.164 | 1.687 | \-0.850 | 0.290 |
>
> Table 6 in the paper. The mean responsiveness scores of the four pairs of models, evaluated using a 7-point comparison category rating. A positive score indicates that the left model (model A) is more responsive than the right one (model B), while a negative number means that model B is more responsive. The larger the absolute value of the score is, the more obvious the responsiveness of the two models is for human evaluators.

---

> > ### Author Response · Authors · 2025-11-20
> > **Response to Reviewer G3Jr (2/2)**
> >
> > > It is unclear how STITCH handles cases where reasoning takes longer than speech generation. Would the model introduce pauses until reasoning is completed? Such pauses might make the users feel more inconvenient to use it.
> >
> > STITCH will **not** introduce a pause when waiting for reasoning that is longer than speech generation. This is because in the training data, the number of speech chunks is always more than the number of reasoning chunks. In the last paragraph of Section 3.2, we highlight that we *remove training samples where the reasoning takes longer than speech generation*. The model will learn not to stop speaking before the reasoning ends. Recall that the model will emit a special token (\[EOR\]) that represents that the reasoning has ended. Before this token is emitted, the speech generation will not pause. During inference, we do not observe STITCH-S and STITCH-R to generate reasoning that is longer than the speech. We added this discussion to Section 5 in our revision.
> >
> > > In STITCH-S, reasoning occurs after the response. In this case, the reasoning could be merely post-hoc justification. Why does this variant outperform baselines that reason before speaking or models without reasoning?
> >
> > We answer the above concern in two parts.
> >
> > #### Can the reasoning be merely post-hoc justification?
> >
> > In Appendix B, we investigate whether reason chunks anticipate future responses or are merely *post-hoc* justifications of previous responses.  We use GPT-4o as a judge, give 4o the full interleaved reasoning-text chunks generated by STITCH-R/S, and ask GPT-4o to determine whether the thinking is only a post-hoc justification of the previous text chunks. We find that in 95% of cases, the reasoning chunk is not the post-hoc explanations of prior response but rather the anticipation of future text chunks.
> >
> > #### Why can STITCH-S outperform baselines while the reasoning occurs after the response?
> >
> > Based on the prior experiment, we have shown that the reasoning chunk in STITCH-S is not a post-hoc explanation of prior responses. The next question is how STITCH generates the first text response chunk without any reasoning. The reason is that the first text chunk tends to rephrase the information in the question without adding new calculation results, as seen in the sample in Table 8 of the Appendix. Since the first text token chunk can be generated without any reasoning, directly generating those texts without conditioning on any reasoning does not harm the performance.
> >
> > Due to limited space, we put the above discussion in Appendix B in the previous version. However, we have moved Appendix B into the main content using the additional 10th page in the revised PDF..
> >
> > We have incorporated the clarifications and revisions above, and we believe they address the concerns raised. If any questions remain, please let us know, and we are happy to provide further details. We are looking forward to learn your feedback on our responses.

---

### Official Review · Reviewer_DbXS · 2025-11-02

**Soundness:** 3
**Presentation:** 3
**Contribution:** 3
**Rating:** 4
**Confidence:** 3

**Summary:**

This paper introduces STITCH, a new framework that allows Spoken Language Models (SLMs) to “think” while they are “talking.” Current SLMs can only generate speech responses directly, without any internal reasoning before speaking. In contrast, humans silently think before responding aloud. STITCH mimics this behavior by dividing the reasoning and response generation into small “chunks.”
Instead of generating a long chain of reasoning first, STITCH alternates between reasoning chunks and speech chunks, so the model can keep thinking while speaking. The paper shows that STITCH improves reasoning performance by 15–20% especially on math tasks compared to baseline models, while keeping response latency nearly the same. It performs equally well on non-reasoning tasks, showing that “thinking” doesn’t harm general speech quality.

**Strengths:**

- The idea is novel — enabling SLMs to think internally while speaking. The chunked reasoning design, STITCH-R and STITCH-S is creative and practical for real-time systems.
- Overall the paper is very well-written

**Weaknesses:**

- The experiments mainly focus on math reasoning datasets like GSM8K and SVAMP. It would be valuable to test STITCH on more diverse reasoning domains such as commonsense, dialogue reasoning, or multi-hop factual reasoning.

**Questions:**

- The paper compares only within GLM-4-Voice variants. Is it possible to see how STITCH performs when applied to other open-source SLMs like Qwen2.5-Omni or Baichuan-Audio to prove generalizability?
- How does STITCH behave under actual streaming conditions where speech synthesis latency varies?

---

> ### Author Response · Authors · 2025-11-20
> **Response to Reviewer DbXS**
>
> >  The experiments mainly focus on math reasoning datasets like GSM8K and SVAMP. It would be valuable to test STITCH on more diverse reasoning domains such as commonsense, dialogue reasoning, or multi-hop factual reasoning.
>
> Because our supervised fine-tuning data focuses on math reasoning, we expect gains primarily in mathematical reasoning. Prior works on fine-tuning LLMs have shown that fine-tuning the models on a specific type of task can significantly boost in-domain performance on that task \[1\], and fine-tuning on diverse domains is critical for the LLM to perform well on diverse domains \[2\]. Consequently, strong performance on other reasoning types generally requires domain-matched supervision during SFT. **As the goal of this paper is to demonstrate that STITCH enables the model to reason without incurring additional latency instead of training a very general model that works for diverse reasoning tasks, we have chosen the most commonly used reasoning task, i.e., mathematical reasoning, as our testbed.**
> Following the reviewer's suggestion, we also include a new experiment on HotPotQA, a multihop factual reasoning dataset. The evaluation settings largely follow our paper. The results are presented in the table below, and we found that all models exhibit a similar performance. Again, **this result should not be interpreted as a weakness of our proposed method; it is because the training data does not include this kind of multihop factual reasoning data instances.**
>
> |  Model | Accuracy |
> | :---- | :---- |
> | GLM-4-Voice | 12.67 |
> | No Reasoning | 10.00 |
> | Mix Reasoning (no) | 10.67 |
> | TBS | 12.33 |
> | Mix Reasoning (yes) | 11.00 |
> | STITCH-R | 10.00 |
> | STITCH-S | 10.33 |
>
> >  The paper compares only within GLM-4-Voice variants. Is it possible to see how STITCH performs when applied to other open-source SLMs like Qwen2.5-Omni or Baichuan-Audio to prove generalizability?
>
> Please kindly refer to the general response to all reviewers.
>
> > How does STITCH behave under actual streaming conditions where speech synthesis latency varies?
>
> As long as the speech synthesis latency $t\_{syn}$ is less than the duration of the corresponding speech chunk $t\_{chunk}$, the speech synthesis latency does not need to be considered. The rationale is that the speech decoder, which generates the audio waveform from the speech tokens, can be run in parallel with the STITCH model, which generates the tokens. We include a new Figure 5 in the updated PDF to illustrate this.
>
> In actual streaming conditions, current speech decoders are always designed to have $t\_{syn}$ less than $t\_{chunk}$, otherwise the speech synthesis speed cannot catch up with the speech playout speed. The above claim is true regardless of whether STITCH is used.
>
> We have incorporated the clarifications and revisions above, and we believe they address the concerns raised. If any questions remain, please let us know, and we are happy to provide further details. We are looking forward to learn your feedback on our responses.
>
>
> --------------
>
> \[1\] Zelikman, Eric, et al. "Star: Bootstrapping reasoning with reasoning." Advances in Neural Information Processing Systems 35 (2022): 15476-15488.
> \[2\] The CoT Collection: Improving Zero-shot and Few-shot Learning of Language Models via Chain-of-Thought Fine-Tuning (Kim et al., EMNLP 2023\)

---

### Author Response · Authors · 2025-11-20
**General Response on Using Other SLMs**

We would like to thank all the reviewers for their valuable feedback. We are glad that reviewers found our paper novel and creative (Review DbXS and TVGk), having good writing and presentation (Reviewer DbXS, G3Jr, Yahr), with comprehensive experiments and analysis on reasoning and non-reasoning tasks (Reviewer G3Jr and TVGk), showing comparable results on reasoning datasets while reducing the latency (Reviewer Yahr), and making the proposed method practical for real-time systems (Reviewer DbXS).

Below, we address one common comment raised by multiple reviewers.

## Is it possible to see the results on other SLMs?

This is indeed a very natural and reasonable question to ask. However, at the time of writing this paper, GLM-4-Voice was the only practical open model available. Below, we provide more details on why we deliberately chose GLM-4-Voice over other SLMs as our model in our experiment. In short, **due to practical limitations of other open-source SLMs, not limitations of our method**, fine-tuning other models would require substantially more compute beyond our budget, as detailed below.

1. **Qwen-2.5-Omni**: Qwen-2.5-Omni is a thinker-talker model, where the thinker takes the speech input and generates the text output, while the talker takes the continuous representation from the thinker model and generates the audio tokens as the output. There is an important reason why we do not fine-tune this model.
   1. **The speech tokenizer of Qwen-2.5-Omni was not released, which prevents fine-tuning the talker**. To fine-tune Qwen-2.5-Omni to operate in speech-to-speech, one needs to prepare the training data that comprises paired speech input and speech output, where the output is in the format of audio tokens. However, the speech tokenizer, *qwen-tts-tokenizer*, which is required to convert the target output speech into its corresponding speech tokens, is not publicly available. This remains an unresolved issue that has been pending on GitHub since April 2025 \[1,2\].

Consequently, fine-tuning Qwen-2.5-Omni for speech-to-speech is challenging because the talker must co-adapt to the thinker’s output hidden state, but the speech tokenizer is not publicly available. Methodologically, applying STITCH to the thinker-talker model is not specifically challenging. We ran a preliminary experiment that only fine-tunes the thinker model of Qwen-2.5-Omni using speech-in-text-out data. The target output is the interleaved reasoning-text (the green and orange blocks in Figure 2 in our paper). We set the interleaving ratio tailored for Qwen-2.5-Omni. **After fine-tuning, the thinker model successfully generates interleaved reasoning-text output**. The next step would be to pass these thinker output hidden states to the talker; however, the talker does not recognize the new hidden-state interface, and without access to the corresponding speech tokenizer, aligning the talker is challenging.

2. **Baichuan Audio**: While this is also an interleaved SLM, **the text-speech interleaved ratio is not fixed.**  The technical report from Baichuan states, “The *interleaved data is segmented using LLM, which performs natural segmentation based on punctuation or natural pauses in the text content.*” STITCH requires the SLM to output text-speech tokens in a fixed ratio. If we use Baichuan-Audio as the base SLM, we will need to train it to generate fixed-ratio text-speech tokens, adding engineering and training overhead unrelated to our contribution.

3. **Moshi**: We did not select it because its performance is weaker than that of many newer open-source SLMs, as evidenced by the performance comparison between Moshi and GLM-4-Voice \[3\]. If we start our experiment from Moshi, we might need to spend a substantial additional compute and data to make Moshi perform equally well as other open-source SLMs, like GLM-4-Voice. As a result, starting from GLM-4-Voice is a more practical choice for our study.

Beyond the limitations of those SLMs themselves, we want to note the **monteray cost of fine-tuning one SLM is prohibitively high.**  Conducting the main experiments shown in Table 1 for a different SLM will require approximately 2,720 GPU hours using A100. This will cost around 3000 USD even if we rent the cheapest GPU on Vast.ai, which is infeasible for us. While we understand that it is natural to request experiments on different architectures, we hope the reviewer can consider the cost of running experiments on various SLMs. As proof of concept, we believe that our paper has demonstrated great potential in the model we use.


--------------

- [1] https://github.com/QwenLM/Qwen2.5-Omni/issues/219
- [2] https://github.com/QwenLM/Qwen2.5-Omni/issues/133
- [3] Zeng, Aohan, et al. "Glm-4-voice: Towards intelligent and human-like end-to-end spoken chatbot." arXiv preprint arXiv:2412.02612 (2024).

---

### Author Response · Authors · 2025-11-20
**Summary of PDF Revision**

The following change log is included to help readers understand the modifications made throughout
the author-reviewer discussion period.

**Nov. 19th**
First revision based on the initial reviews from the reviewer. The modifications are highlighted in **blue**. The main differences are summarized as follows:

1. **Explain the difficulty of fine-tuning Qwen-2.5-Omni:**
   In Section 3, we add a sentence to explain why we do not fine-tune Qwen-2.5-Omni.

2. **Statistical tests in Section 5:**
   We run McNemar’s test to see if the performance between no-reasoning baselines and other models is statistically significant and report the results in Section 5.

3. **Explaining that reasoning always finishes before text-speech segments end:**
   In Section 3, we include a paragraph explaining that the reasoning segments generated by STITCH will always be less than the text/speech segments.

4. **Human evaluation in Section 5:**
   We add a paragraph in Section 5, where we run a human evaluation comparing the responsiveness of different pairs of models.

5. **Explain when to use STITCH-R/STITCH-S:**
   Based on all the previous experiment results, in the last paragraph in Section 5, we explain when to use STITCH-R and STITCH-S.

6. **Qualitative results moved to Section 6:**
   We move the qualitative results into the main content, where we illustrate how the reasoning of STITCH-R and STITCH-S precedes the text responses most of the time.

7. **Add Figure 5:**
   We add Figure 5 to show that different $t_{syn}$ will not break the thinking while speaking pattern of STITCH as long as \$t_{syn}<t_{chunk}\$.

---

### Author Response · Authors · 2025-11-26
**Gentle Reminder of  Author-Reviewer Discussion Period**

Dear reviewers,

If you have already read our rebuttal and made the necessary acknowledgement as specified in the email to the reviewers, please omit our message. If you haven't read our responses and made the necessary changes in the original reviews and scores, this is a gentle reminder that there is less than one week for you to interact with the authors. We have provided detailed and comprehensive responses addressing the weaknesses and questions you raised in the original review, and we look forward to hearing back from you.

Thank you very much.

Below is quoted from the email to the reviewer

>At this point authors have posted responses to the concerns you raised in your review. If you have not done so already, we ask that you read through these posted responses and then post a reply (a) acknowledging the rebuttal, and (b) **precisely identifying which of your concerns have been addressed, which have not been addressed and why**. Please also **update your review score if your concerns have been addressed**.
Please do so within the next week: the absolute last date for you to respond to author comments is December 2.

---

### Author Response · Authors · 2025-11-30
**Summary of Review and Rebuttal for AC (1/3)**

We faithfully summarize the strengths and weaknesses listed by the reviewer, and we elaborate on how we mitigate the weaknesses mentioned by each reviewer. We also include a brief summary of the discussion during the discussion period before it was cut off abruptly. For the sake of simplicity, we only summarize our brief response and combine multiple weaknesses from different reviewers when they are similar. The point-to-point full response can be found in individual replies to the reviewer. **We properly address every weakness and question raised by every reviewer.**

## **Summary of Strength**

The reviewers found our paper novel and creative (DbXS and TVGk), having good writing and presentation (DbXS, G3Jr, Yahr), with comprehensive experiments and analysis on reasoning and non-reasoning tasks (G3Jr and TVGk), showing comparable results on reasoning datasets while reducing the latency (Yahr), and making the proposed method practical for real-time systems (DbXS), and clearly showing the effectiveness of the proposed method (TVGk). The investigation into optimal reasoning chunk length and the use of reasoning from other models adds further novelties (TVGk).

## **Summary of Weaknesses, Questions, and How We Address Them**

### W1. Missing User-Centric Evaluation of Responsiveness/User Experience (G3Jr)

> The main motivation for reducing interaction latency is to improve user experience while maintaining response quality, but the paper does not include human evaluation.

**Our response: We add human evaluation** and show that humans find our proposed method, STITCH, to have better responsiveness compared to other baselines, with STITCH-S being the most responsive. The experiment has been included into the revision.

 ---
### W2. Lack of Clear Guidelines for STITCH-R vs STITCH-S (G3Jr)

> There are no clear guidelines on when to use STITCH-R versus STITCH-S.

**Our response**: In Section 5 of our revision, **we add a paragraph** to explain the accuracy-latency trade-off between STITCH-R and STITCH-S, providing clear guidelines for users.

 ---
### W3. The results of Table 1 Do Not Have a Clear Winner. The paper does not clearly explain the reasons behind these trends. (G3Jr)

> Based on the performance tables (Table 1-(a) and 1-(b)), there is no clear winner that consistently outperforms all other baselines. The paper does not clearly explain the reasons behind these trends.

**Our response:** Relevant discussions and explanations are already presented in Section 5. The performance in Table 1(a) and (b) is not used to show which model is the best, but rather to show the trade-off between performance and latency.

 ---
### W4. It is unclear how STITCH handles reasoning that is longer than speech generation. (Yahr, G3Jr)

> The paper does not provide an analysis or guarantee as to whether the reasoning path consistently concludes earlier in the 100 / 13 / 26. (If the reasoning is longer than the speech), would the model introduce pauses until reasoning is completed?

**Our response:** We explain that due to how we design the training data, the situation described by the reviewer **does not occur** in the training data. During inference time, the problematic behavior mentioned by the reviewer also **never happens**. **We added a paragraph in Section 5 in the revision to discuss this**.

 ---
### W5. The scope of its novelty and generalizability appears somewhat limited. (Yahr)

> The interleaving method applied in this study appears to extend the existing text–speech interleaving approach in prior SLMs.

**Our response:** No prior work introduces reasoning to SLMs, and this is already a significant novelty. The prior works mentioned by the reviewer all interleave spoken contents, while STITCH interleaves unspoken reasoning and spoken contents. This separation between “silent” reasoning and spoken content is a core conceptual difference from prior interleaved SLMs. STITCH is the first demonstration that reasoning can be integrated into SLMs while preserving streaming latency

 ---
### W6. The paper does not experiment with additional SLM architectures. (Yahr, DbXS)

> It remains uncertain whether the proposed approach would generalize well to alternative architectures, such as those that jointly optimize a separate decoder (or talker) during inference rather than interleaving, or models that predict codecs (e.g., moshi).

**Our response:** In our paper, we explicitly mention that the proposed method can be extended to thinker-talker SLMs, while we do not use a thinker-talker SLM, e.g., Qwen-2.5-Omni, since **the speech tokenizer of Qwen-2.5-Omni is not open-sourced, making the speech-to-speech fine-tuning almost impossible**. This is mentioned in multiple GitHub issues on the official repository. Our preliminary experiments on the thinker of Qwen-2.5-Omni show that STITCH is applicable to the thinker-talker model. **We do not use Moshi since the base model’s performance is weaker than the GLM-4-Voice we use.**

---

> ### Author Response · Authors · 2025-11-30
> **Summary of Review and Rebuttal for AC (2/3)**
>
> ### W7. The paper only uses A100+vLLM for inference, limiting the applicability. (Yahr)
>
> > The reported optimization is based on the A100 \+ vLLM setting, which may limit the applicability of the results. It remains uncertain whether the proposed approach would generalize well to limited hardwares.
>
> **Our response: Our analysis in Section 7 considers the case when the hardware has a smaller token throughput.** The results in Section 7 (and Figure 3(a) in the appendix) show that with slower hardware, STITCH can still perform reasonably well most of the time. We also mention that accelerating inference speed is a more engineering problem, and possible speed-up methods, including speculative decoding or quantization, can be used.
>
> ---
>
> ### W8. The paper does not discuss different choices of models for generating the reasoning path in the dataset. (Yahr)
>
> > Does the choice of path-generating model (e.g., GPT-4o vs. reasoning-oriented models) materially affect downstream performance, **especially on non-reasoning tasks**?
>
> **Our response**: We have also tried to use Deepseek-r1 to generate the reasoning path, but **we do not find the resulting reasoning paths to be different from those generated by GPT-4o**. Eventually, we use GPT-4o since it is better on non-reasoning tasks, easier to use, and more controllable.
>
> ---
> ### W9. STITCH-S generates a speech chunk before the first reasoning chunk. Does this make the reasoning merely a post-hoc justification of previous speech? (Yahr, G3Jr)
>
> **Our response:** We investigate whether reason chunks anticipate future responses or are merely post-hoc justifications of previous responses by a well-designed experiment. We find that **in 95% of cases, the reasoning chunk is not the post-hoc explanations of prior response but rather the anticipation of future text chunks. The experiment is originally located in the appendix and is not moved to the main content.**
>
> ---
> ### W10. The paper does not compare STITCH against thinker-talker models. (TVGk)
>
> > Models like Qwen2.5-Omni should serve as the no-reasoning baseline in the benchmark comparison. Since thinker-talker models address the limitations of the interleaved models, they should at least serve as no-reasoning baselines to ensure fair comparisons.
>
> **Our response**: Using Qwen-2.5-Omni may not be a reasonable baseline because the Qwen-2.5-Omni has undergone a supervised fine-tuning (SFT) stage that might deliberately include a math reasoning dataset. On the other hand, the model with use, GLM-4-Voice, deliberately excludes math datasets during SFT. Therefore, it is unclear whether the thinker-talker models, as a modeling architecture, truly address the limitation of the interleaved models, or whether they are simply trained on different SFT datasets. **Comparing Qwen-2.5-Omni with the models trained from GLM-4-Voice in our paper cannot help us reach a meaningful conclusion since they have different training data.**
>
> ---
> ### W11. Evaluation is conducted solely on text outputs. (TVGk)
>
> > It would be more robust to also evaluate directly on speech outputs, since the performance gap between text and speech evaluations can depend on the backbone LLM’s text-speech alignment capabilities.
>
> **Our response**: **As mentioned in Section 4.2, our paper evaluates both the text and speech output.** Precisely, we run ASR to get the transcription of the speech output and then evaluate the text transcription. The results, as shown in Table 6 in the Appendix, reveal that all observations we found in the text outputs are consistent with the results on the speech output. However, the speech output exhibits a slight performance drop of around 3% to 6%, which may be attributed to imperfect ASR transcription.
>
> ---
> ### W12. There is an inconsistency in the introduction section. (TVGk)
>
> **Our response:** We have tweaked the phrasing to avoid misunderstanding.
>
> ---
> ### W13. Statistical analysis should be performed to assess whether there are significant differences in responses among TBS, STITCH-R, STITCH-S, etc. (TVGk)
>
> **Our response**: We add McNemar’s test (two-sided,  $\\alpha=0.05$), which is to justify our claims in Section 5 to say one model is significantly better than another. We added this to the revision.
>
> ---
> ### W14. The experiments mainly focus on math reasoning datasets like GSM8K and SVAMP.**  **(DbXS)
>
> > It would be valuable to test STITCH on more diverse reasoning domains such as commonsense, dialogue reasoning, or multi-hop factual reasoning.
>
> **Our response:** As the goal of this paper is to demonstrate that STITCH, as a generation method, enables the model to reason without incurring additional latency. Our goal is not to train a very general model that works for diverse reasoning tasks. Consequently, we have chosen the most commonly used reasoning task, i.e., mathematical reasoning, as our testbed.

---

> ### Author Response · Authors · 2025-11-30
> **Summary of Review and Rebuttal for AC (3/3)**
>
> ### Q1. How does STITCH behave under actual streaming conditions where speech synthesis latency varies? (DbXS)
>
> **Our response:** STITCH behaves as well as other TTS models in streaming conditions. We elaborate on why this is the case and add a figure in the revision to explain this.

---

### Author Response · Authors · 2025-12-04
**Summary of Author-Reviewer Discussion Period**

## **Summary of the Discussion Period**

1. Reviewer Yahr: On Nov. 20th and 21st, the reviewer engaged in a multi-turn discussion. Eventually, **the reviewer raised the score from 4 (weak reject) to 6 (weak accept) on Nov. 21st, far before the widespread OpenReview incident.**
2. Reviewer TVGk: On Nov. 23, the reviewer added a follow-up clarification, and we responded to them immediately. However, we do not hear the feedback from the reviewer, and the discussion period was cut off.
3. Reviewer DbXS and G3Jr: We did not get the responses from the two reviewers before the discussion period was cut off.

Since the remaining weaknesses and questions from the reviewers, except Reviewer Yahr (who already raised the score), are objective questions about the experiment settings and methodology, and we believe our answers, additional experiments (human evaluation, statistical tests, and qualitative analysis), and revision address those weaknesses. **The author is confident that the remaining reviewer will acknowledge that the rebuttal largely mitigates their original concern and decide to increase the rating based on our responses, should the discussion period not be cut off.**

---

### Meta-Review · Area_Chair_jDho · 2026-01-06

**Summary:**

This paper present a smart trick that having the model to think while speaking as speaking is slower than the LLM token generation rate.

The concerns are mainly on
- novelty and contribution is limited
- if only math get improved
- if interleave reasoning and speaking is a good design
- if low latency leads to better experience

**Reviewer Concerns:**

- novelty and contribution is limited

Addressed (my personal opinion on this is this paper is defintely novel, the core idea is very/too simple, but the contribution might be large as it is flexible and can be deployed to any model; i am comparing this paper with the CoT paper, which is also extremely simple but now became the standard).

- if only math get improved

Addressed by additional experiment

- if interleave reasoning and speaking is a good design

Addressed by demonstrating the reasoning is not post-hoc

- if low latency leads to better experience

Addressed by human eval

**Reviewer Scores:**

Yahr: 4 → 6 (reviewer mentioned increasing score to 6)

G3Jr: 6 → 8
because the human eval was added.

TVGk: 6 → 8
because the concern about interleave reasoning was addressed

DbXS: 4 → 6
because the concern about generalizability, streaming behavior, and evaluation methodology were addressed

---

### Decision · Program_Chairs · 2026-01-26

Accept (Poster)